# Nuclear lactate dehydrogenase A senses ROS to produce α-hydroxybutyrate for HPV-induced cervical tumor growth

Yuan Liu[1], Ji-Zheng Guo[2,3], Ying Liu[4], Kui Wang[1], Wencheng Ding[5], Hui Wang[5], Xiang Liu[6], Shengtao Zhou[7], Xiao-Chen Lu[2], Hong-Bin Yang[2], Chenyue Xu[4], Wei Gao[1], Li Zhou[1], Yi-Ping Wang[2], Weiguo Hu [8], Yuquan Wei[9], Canhua Huang[1] & Qun-Ying Lei [2,10]

It is well known that high-risk human papilloma virus (HR-HPV) infection is strongly associated with cervical cancer and E7 was identified as one of the key initiators in HPV-mediated carcinogenesis. Here we show that lactate dehydrogenase A (LDHA) preferably locates in the nucleus in HPV16-positive cervical tumors due to E7-induced intracellular reactive oxygen species (ROS) accumulation. Surprisingly, nuclear LDHA gains a non-canonical enzyme activity to produce α-hydroxybutyrate and triggers DOT1L (disruptor of telomeric silencing 1-like)-mediated histone H3K79 hypermethylation, resulting in the activation of antioxidant responses and Wnt signaling pathway. Furthermore, HPV16 *E7* knocking-out reduces LDHA nuclear translocation and H3K79 tri-methylation in K14-HPV16 transgenic mouse model. HPV16 E7 level is significantly positively correlated with nuclear LDHA and H3K79 tri-methylation in cervical cancer. Collectively, our findings uncover a non-canonical enzyme activity of nuclear LDHA to epigenetically control cellular redox balance and cell proliferation facilitating HPV-induced cervical cancer development.

[1] Department of Biotherapy, State Key Laboratory of Biotherapy and Cancer Center, West China Hospital, and West China School of Basic Medical Sciences & Forensic Medicine, Sichuan University, and Collaborative Innovation Center for Biotherapy, Chengdu 610041, P.R. China. [2] Fudan University Shanghai Cancer Center and Cancer Metabolism Laboratory, Institutes of Biomedical Sciences, Shanghai Medical College, Fudan University, Shanghai 200032, P.R. China. [3] Department of Biochemistry and Molecular Biology, School of Basic Medical Science, Shanghai Medical College, Fudan University, Shanghai 200032, P.R. China. [4] Department of Pathology, School of Basic Medical Sciences, Shanghai Medical College, Fudan University, Shanghai 200032, P.R. China. [5] Department of Obstetrics and Gynecology, Tongji Hospital, Tongji Medical College, Huazhong University of Science and Technology, Wuhan 430030, P.R. China. [6] Department of Pathology, Sichuan Academy of Medical Sciences, Sichuan Provincial People's Hospital, Chengdu 610072, P.R. China. [7] Department of Gynecology and Obstetrics, Key Laboratory of Obstetrics and Gynecologic and Pediatric Diseases and Birth Defects of Ministry of Education, West China Second Hospital, Sichuan University, Chengdu 610041, P. R. China. [8] Fudan University Shanghai Cancer Center and Institutes of Biomedical Sciences, Shanghai Medical College, Department of Oncology, Shanghai Medical College, Fudan University, Shanghai 200032, P.R. China. [9] Department of Biotherapy, State Key Laboratory of Biotherapy and Cancer Center/Collaborative Innovation Center for Biotherapy, West China Hospital, Sichuan University, Chengdu 610041, P.R. China. [10] State Key Laboratory of Medical Neurobiology, Fudan University, Shanghai 200032, P.R. China. These authors contribute equally: Yuan Liu, Ji-Zheng Guo, Ying Liu. Correspondence and requests for materials should be addressed to C.H. (email: hcanhua@scu.edu.cn) or to Q.-Y.L. (email: qlei@fudan.edu.cn)

Cervical cancer is the third most common cancer in women worldwide with about 528,000 new cases and 266,000 deaths annually[1]. Among those, about 95% cases are caused by persistent infections with HR-HPVs[2]. During high-risk HPV infection, two viral early genes, *E6* and *E7*, were identified to play key roles in carcinogenesis by regulating signaling pathways related to cellular transformation[3,4]. Moreover, recent studies reported that HPV16 E7 was the more potent driver for cervical cancer[5–9], and elevation of HPV16 E7 was required to sustain a malignant phenotype in primary cervical cancer[10]. HPV16 E7 has been shown as an essential factor for viral replication in human keratinocytes[11]. And yet, the new mechanism of E7 protein on HPV-induced cervical carcinogenesis remains to be discovered.

Cellular ROS are elevated during the process of many virus infections to host cells, including Epstein–Barr virus (EBV)[12], hepatitis C virus (HCV)[13], hepatitis B virus (HBV)[14], and HPVs[15,16]. The virus-induced chronic oxidative stress increases the integration frequency of HPV16 DNA in human keratinocytes[17]. However, excessive ROS levels are fatal to host cells. It is necessary to increase the antioxidant capacity to counteract with the excessive ROS. In response to oxidative stress, nuclear factor erythroid-derived 2-like 2 (NRF2), a key regulator of cellular antioxidant responses, plays an important role in antioxidant defense through transcriptional activation of antioxidant-response-element (ARE)-bearing genes[18,19]. However, it remains unclear how HPVs control the highly elevated intracellular ROS levels.

Altered cell metabolism is one of the hallmarks of cancer[20]. Metabolism is subtly orchestrated in cancer cells, one frequent phenomenon being a shift from respiration to aerobic glycolysis (known as "Warburg effect")[21]. Early studies suggested that HPV16 E7 oncoprotein affected cellular metabolism on transcription levels[22]. Moreover, inhibition of hexokinase 2 (HK2), the first rate-limiting enzyme of glycolysis, contributed to suppressing HPV16 E7-induced glycolytic phenotype of tumor metabolism and inhibited tumor cell growth[23]. Enhanced expression of LDHA, which is needed for maintaining glycolysis and other metabolic activities, has been associated with the evolution of aggressive and metastatic cancers in a variety of tumor types, including cervical cancer[24]. Silencing of LDHA significantly attenuated colony-formation ability and invasive capacity of cervical cancer cells[25]. Here, we report that LDHA translocates into the nucleus in response to HPV16/18 E7, which depends on elevated ROS levels, and gains a noncanonical enzyme activity to produce an antioxidant metabolite, α-hydroxybutyrate (α-HB). The accumulation of α-HB further activates antioxidant response and Wnt signaling through epigenetic modifications. We present a molecular mechanism that explains how HR-HPVs promote cervical cancer development through nuclear translocation of LDHA and the product of noncanonical enzyme activity, α-HB, providing a crosstalk between cell metabolism and epigenetic signaling networks.

## Results

**HPV-induced LDHA nuclear translocation is dependent on ROS.** HPV infection is the most common factor of cervical cancer. Among all HR-HPVs, HPV16 is the most carcinogenic HPV type and accounts for ~55%–60% of all cervical cancers, and HPV18 is the second type which accounts for 10%–15%[26]. Persistent oxidative stress is always accompanied by the process of HPV infection to host cells[15,16]. The altered cellular redox levels may cause reversible modifications on specific cysteine residues and affect the protein functions[27–29]. To decipher the changes in the cysteine proteome on HR-HPV infection, we developed a sensitive and specific redox proteomics method using iodoacetyl

tandem mass tag (iodoTMT) reagents composed of a sulfhydryl-reactive iodoacetyl group selective labeling sulfhydryl (-SH) groups and sets of isobaric isomers which can be differentiated by mass spectrometry (MS), enabling quantitation of the relative abundance of cysteine modifications (Supplementary Fig. 1a). To identify HPV-related redox-sensitive effectors, the cysteine proteomes were obtained from C33A (HPV negative), SiHa (containing HPV16 genome), and HeLa (containing HPV18 genome) cells (Supplementary Fig. 1b). The key glycolysis enzyme LDHA was identified to be a potential key regulator in HPV-induced cervical cancer development (Supplementary Data 1).

To investigate the potential role of LDHA in HPV-induced cervical cancer, we collected HPV-negative and HPV-positive cervical cancer specimens from 66 patients (Supplementary Data 2). Immunohistochemistry (IHC) showed that LDHA nuclear staining was significantly increased in HPV16-positive tumors compared with HPV16-negative tumors (Fig. 1a, b). To test whether the pivotal oncoprotein HPV16/18 E7-mediated LDHA nuclear translocation, we packaged retrovirus containing HPV16/18 *E7* gene and infected primary human cervix keratinocytes (PHKs), immortalized human keratinocyte cell line HaCaT, and transfected HPV16 *E7* gene into HPV-negative human cervical cancer cell line HT-3 (Supplementary Fig. 2a). As expected, HPV16/18 E7 expression dramatically increased the percentage of LDHA nuclear-translocated cells from ~5% to ~50% (Fig. 1c, d, and Supplementary Fig. 2b, c). In line with the potential effect of HPV infection on ROS production, we found that HPV16/18 E7 induction resulted in cellular ROS accumulation (Fig. 1e and Supplementary Fig. 2d). Notably, supplement with a ROS scavenger N-acetyl-L-cysteine (NAC) remarkably reduced LDHA nuclear translocation in HPV16/18 E7-transduced cells (Fig. 1c, d, and Supplementary Fig. 2b, c). This observation triggered us to speculate that ROS possibly promote LDHA nuclear translocation. To this end, we treated HaCaT, HT-3, U2OS, and HeLa cells with hydrogen peroxide ($H_2O_2$) and found that LDHA rapidly translocated from the cytoplasm to nuclear in a dose-dependent manner, and the $H_2O_2$-induced subcellular redistribution of LDHA was reversed by NAC supplement (Fig. 1f, g, and Supplementary Fig. 3a–d). Meanwhile, the cellular ROS levels were measured upon $H_2O_2$ and NAC treatment in HT-3 and U2OS cells under the same condition (Supplementary Fig. 3e). To further validate this, we performed nuclear isolation assay and found the similar pattern for LDHA localization (Fig. 1h). These data indicated that LDHA nuclear translocation induced by HPV infection is dependent on ROS.

Mitochondrial respiratory chain and nicotinamide adenine dinucleotide phosphate (NADPH) oxidases (NOXs) contribute the major source of endogenous ROS in mammals[30,31]. To figure out the potential source of ROS induced by HPV16/18 E7, mitochondrial and NOXs originated ROS were measured, respectively. As shown in Supplementary Fig. 4a, HPV16/18 E7 expression increased mitochondrial ROS generation, as determined using the fluorescent dye MitoSOX Red. On the other hand, NOXs inhibitor diphenyleneiodonium chloride (DPIC) decreased the intracellular ROS level (Supplementary Fig. 4b), suggesting that NOXs contributes to HPV16/18 E7-induced intracellular ROS elevation as well as mitochondrial respiratory chain.

**Nuclear LDHA gains a noncanonical activity to produce α-HB.** Lactate dehydrogenase acts an indispensable role in cancer development and possesses a well-defined canonical enzyme activity, catalyzing the conversion between pyruvate (Pyr) and lactate (Lac)[32]. It is well known that LDHA localizes at cytoplasm exerting its lactate-producing activity. Interestingly, LDHA has

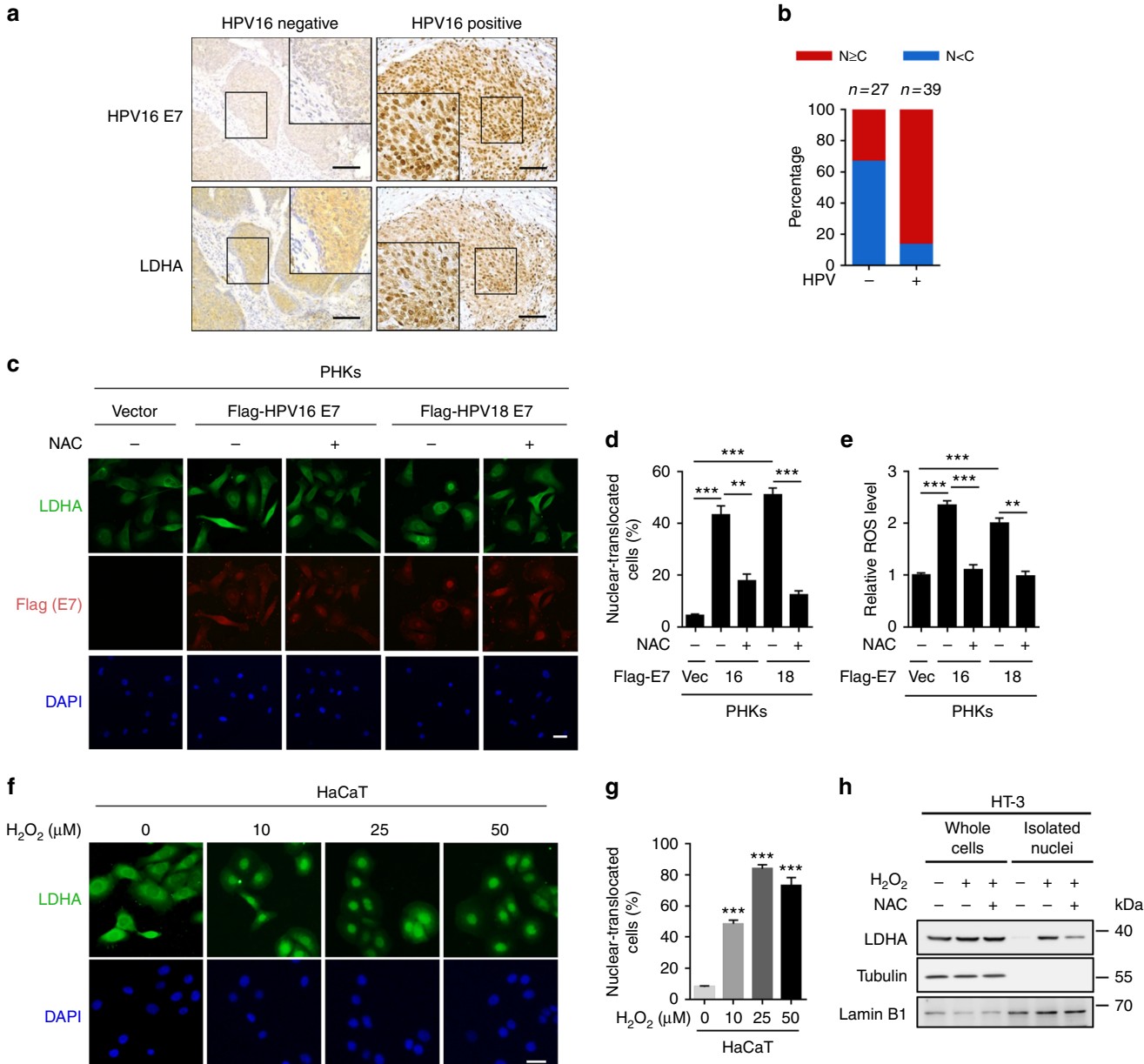

**Fig. 1** HPV16/18 E7 induces LDHA nuclear translocation by ROS accumulation. **a** LDHA is significantly translocated into nucleus in HPV16 positive cervical cancer tissues. Representative IHC images for LDHA localization in HPV16-negative and positive cervical tumor samples. Scale bar, 100 μm. **b** Nuclear LDHA is dramatically increased in HPV16-positive cervical cancer tissues. Semi-quantitative cytoplasmic LDHA and nuclear LDHA scoring was performed in HPV16 negative ($n = 27$) and positive ($n = 39$) cervical tumor samples. **c**, **d** HPV16/18 E7 promotes LDHA nuclear translocation in a ROS-dependent manner. Primary human cervix keratinocytes (PHKs) stably expressing vector or Flag-tagged HPV16/18 E7 were treated with or without 1 mM NAC for 6 h, followed by staining with anti-LDHA (green), anti-Flag (red) antibodies, and DAPI (blue). Scale bars, 10 μm (**c**). The percentage of cells with nucleus-localized LDHA compared to total cell number was quantified (**d**). **e** HPV16/18 E7 enhances ROS production. Cellular ROS were measured in PHKs stably expressing vector or HPV16/18 E7 coupled with or without 1 mM NAC treatment for 6 h, followed by using the ROS-sensitive fluorescent dye (CM-$H_2$DCFDA) with flow cytometry according to the manufacturer's protocol. **f**, **g** LDHA nuclear translocation is profoundly increased in an $H_2O_2$-dose-dependent manner. Immunofluorescent images of LDHA (green) in HaCaT cells upon different dose of $H_2O_2$ treatment as indicated. DAPI, blue. Scale bars, 10 μm (**f**). The percentage of cells with nucleus-localized LDHA compared with total cell number was quantified (**g**). **h** ROS promote LDHA nuclear translocation. HT-3 cells were treated with or without 10 μM $H_2O_2$ for 6 h, and supplemented with or without 1 mM NAC for extended 6 h as indicated for nuclear isolating, followed by blotting with LDHA, Tubulin, and Lamin B1. Results are representative of three independent experiments. All data are shown as mean ± SEM. The $p$ values were determined by two-tailed $t$-test. The values of $p < 0.05$ were considered statistically significant. *, **, and *** denote $p < 0.05$, $p < 0.01$, and $p < 0.001$, respectively. NS means non significant

also been shown to exhibit noncanonical enzyme activities[33–35]. We hypothesized that, after nuclear translocation, LDHA probably gained a noncanonical enzyme activity. To this end, we discovered that LDHA acquired a α-HB-producing activity in the nucleus (Fig. 2a and Supplementary Fig. 5). We measured both the canonical and noncanonical enzyme activities upon $H_2O_2$ exposure or HPV16 E7 induction, with or without NAC supplement. Interestingly, the noncanonical enzyme activity but not

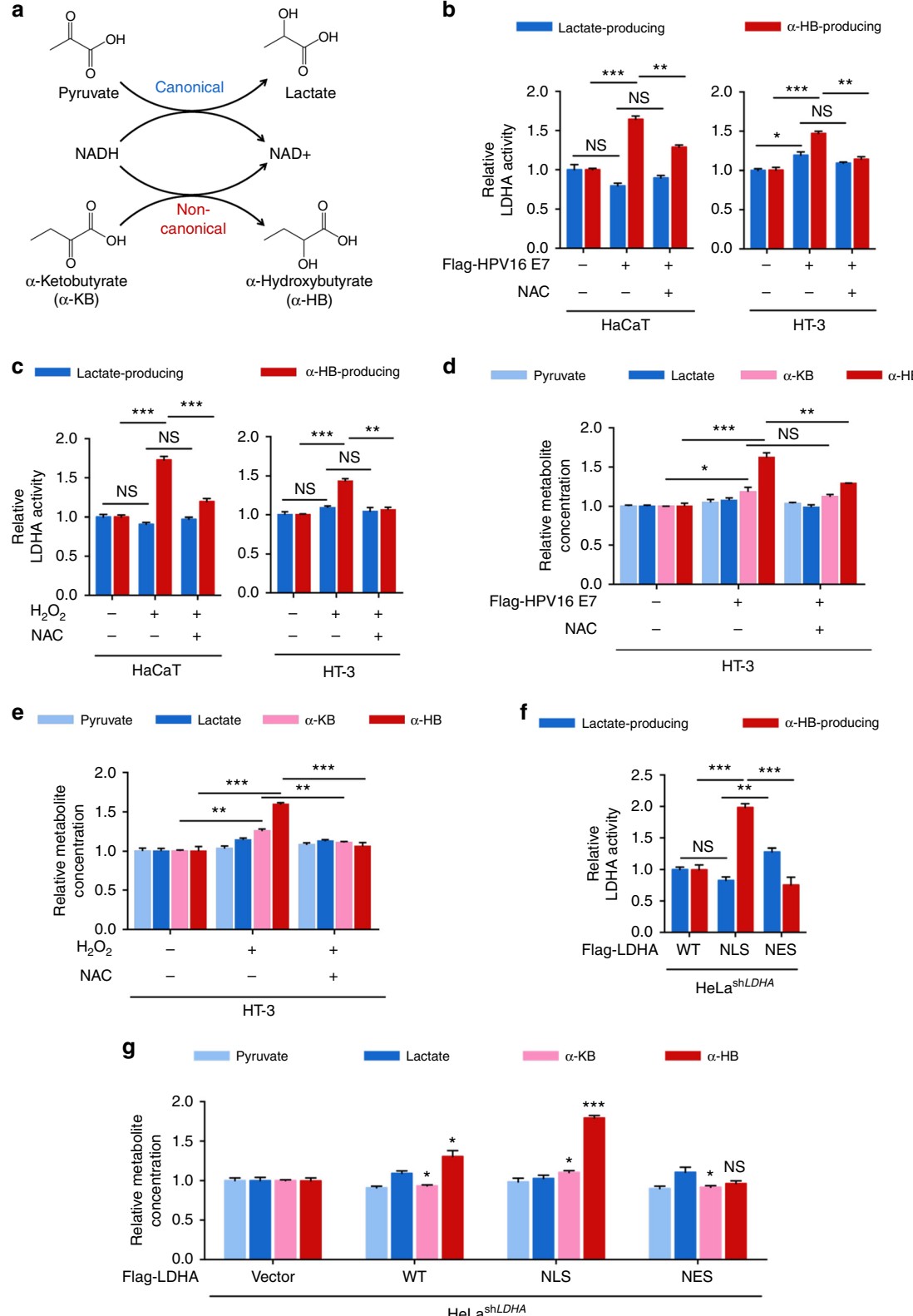

the canonical activity of LDHA was elevated by ~1.5-fold after HPV16 E7 induction or $H_2O_2$ treatment (Fig. 2b, c). Moreover, NAC supplement decreased noncanonical enzyme activity of LDHA, suggesting a ROS-dependent switch in LDHA activity. In addition, we quantified four direct substrates of LDHA, including pyruvate, lactate, α-ketobutyrate (α-KB), and α-HB, using liquid chromatography coupled to triple quadrupole tandem mass

spectrometry (LC-MS/MS). As expected, α-HB, the product of noncanonical LDHA activity, accumulated by ~1.6-fold after HPV16/18 E7 induction or $H_2O_2$ treatment, while no significant changes were observed on pyruvate or lactate levels (Fig. 2d, e, and Supplementary Fig. 6a–e). Notably, NAC supplement totally blocked the elevation of α-HB (Fig. 2d, e, and Supplementary Fig. 6d, e). To assess the function of nuclear LDHA, we isolated

**Fig. 2** Nuclear LDHA gains a noncanonical enzyme activity to produce α-HB. **a** Schematic of the canonical and noncanonical LDHA enzyme activity. **b** HPV16 E7 enhances the noncanonical enzyme activity of LDHA. The canonical and noncanonical LDHA enzyme activities were measured in HaCaT and HT-3 cells stably expressing vector or HPV16 E7 coupled with or without 1 mM NAC treatment for 6 h. **c** ROS elevate the noncanonical enzyme activity of LDHA. HaCaT and HT-3 cells were treated with or without 10 μM $H_2O_2$ for 6 h, and supplemented with or without 1 mM NAC for extended 6 h as indicated for measurement of the canonical and noncanonical LDHA enzyme activities. **d** HPV16 E7 expression accumulates cellular α-HB. The extracted metabolite samples from the HT-3 cells stably expressing vector or HPV16 E7 coupled with or without 1 mM NAC treatment for 6 h as indicated were analyzed by LC-MS/MS, relative abundance (by metabolite peak area) was shown. **e** ROS accumulate cellular α-HB. The extracted metabolite samples from HT-3 cells treated with or without 10 μM $H_2O_2$ for 6 h, and supplemented with or without 1 mM NAC for extended 6 h as indicated were analyzed by liquid chromatography coupled to triple quadrupole tandem mass spectrometry (LC-MS/MS), relative abundance (by metabolite peak area) was shown. **f** Nuclear LDHA presents higher noncanonical enzyme activity. The canonical and noncanonical enzyme activities were measured in HeLa stable cells with *LDHA* knockdown and Vec/WT/NLS/NES rescue. Vec, vector; WT, wild-type; NLS, nuclear localization signal; NES, nuclear export signal. **g** LDHA nuclear translocation accumulates cellular α-HB. The extracted metabolite samples from HeLa stable cells with *LDHA* knockdown and Vec/WT/NLS/NES rescue were analyzed by LC-MS/MS, relative abundance (by metabolite peak area) was shown. LDHA enzyme activities were normalized to LDHA protein level. Relative metabolite abundances were normalized to cell number. Results are representative of three independent experiments. All data are shown as mean ± SEM. The *p* values were determined by two-tailed *t*-test. The values of $p < 0.05$ were considered statistically significant. *, **, and *** denote $p < 0.05$, $p < 0.01$, and $p < 0.001$, respectively. NS means non significant

nuclei using a well-established, nuclei-specific high-sucrose gradient centrifugation protocol for metabolite measurement (Supplementary Fig. 6b)[36]. Consistent with whole-cell data above, nuclear α-HB was also accumulated upon HPV16/18 E7 induction and $H_2O_2$ treatment while NAC supplement eliminated this induction in HaCaT and HT-3 cells (Supplementary Fig. 6c–e). Furthermore, the accumulation of α-HB was diminished in *LDHA*-knockout (KO) cells (Supplementary Fig. 6f, g), indicating that the noncanonical enzyme activity of LDHA mediates its response to ROS via producing α-HB.

To further test whether the upregulation of noncanonical enzyme activity was a consequence of LDHA nuclear translocation, we generated stable cells with endogenous *LDHA* knockdown and putting back with sh*LDHA* resistant flag-tagged vector, wild-type LDHA (WT) and its mutants containing nuclear localization signal (LDHA[NLS]) and nuclear export signal (LDHA[NES]) peptides, respectively[37] (Supplementary Fig. 7). Consistently, both elevated noncanonical LDHA enzyme activity and α-HB accumulation were observed in LDHA[NLS] stable cells (Fig. 2f, g). Taken together, these data demonstrate that nuclear LDHA gains a noncanonical enzyme activity, leading to accumulation of α-HB.

**ROS disrupt LDHA tetramer to promote noncanonical activity**. To examine whether the LDHA nuclear translocation was associated with LDHA oligomerization, protein crosslinking assay and gel filtration were performed. LDHA tetramers were dramatically decreased by $H_2O_2$ treatment, accompanied by increased dimer and monomer (Fig. 3a). Along with the expression of HPV16 E7 increased LDHA dimer to ~1.9-fold in HaCaT cells and ~1.5-fold in HT-3 cells, respectively (Supplementary Fig. 8a). Next, we fractionated cell extracts by gel filtration and determined the distribution of endogenous LDHA by western blotting. LDHA distributed broadly in multiple fractions but fewer portions were found in dimeric fractions (as determined using protein standards shown in Supplementary Fig. 8b) under normal condition (Fig. 3b and Supplementary Fig. 8c). Upon $H_2O_2$ treatment, LDHA significantly shifted into dimer fractions (Fig. 3b, lower panel). As the oligomerization state of metabolic enzymes closely links to their catalytic activity[38–40], we measured both canonical and noncanonical enzyme activities of tetrameric and dimeric fractions of LDHA. Strikingly, the noncanonical enzyme activity of dimer fractions was significantly increased by more than 1.7-folds while canonical enzyme activity decreased > 3.8-folds compared with that of tetrameric fraction (Fig. 3c). Furthermore, LDHA[NLS] group presented much more dimer fractions than LDHA[NES] group (Fig. 3d, e, and

Supplementary Fig. 8d, e). Taken together, these results demonstrate that ROS-induced nuclear translocation of LDHA, accompanied with its tetramer-to-dimer transition, confers LDHA with a noncanonical enzyme activity.

**α-HB accumulation induces H3K79 hypermethylation**. Aberrant metabolic alterations have been widely associated with tumorigenesis, while the landscape of histone methylation is remodeled with altered metabolism[41,42]. To explore the biological function of intracellular α-HB accumulation, we examined the dimethylation levels of six common histone methylation markers, H3K4, H3K9, H3K27, H3K36, H3K79, and H4K20, in HaCaT cells upon α-HB treatment. Histone H3K79 methylation levels were shown to be upregulated by α-HB in a dose and time-dependent manner (Supplementary Fig. 9a). To confirm the specific effect of α-HB on H3K79 methylation, mono-, di-, trimethylation levels were determined after cells treated with four LDHA-related metabolites in our study (α-HB, α-KB, Pyr, and Lac). Notably, α-HB caused a significant upregulation of histone H3K79 methylation (Fig. 4a). We further determined whether the impact of $H_2O_2$ treatment or HPV16/18 E7 transduction on H3K79 hypermethylation. As shown in Fig. 4b, c, and Supplementary Fig. 9b, histone H3K79 trimethylation level was significantly increased upon HPV16/18 E7 induction or $H_2O_2$ treatment, which was partially reversed by NAC. We next defined whether H3K79 hypermethylation was dependent on LDHA nuclear translocation by using LDHA mutants. Cells expressing LDHA[NLS], but not LDHA[NES], showed elevated H3K79 trimethylation level (Fig. 4d). More importantly, supplementation of α-HB to LDHA[NES] cells remarkably recovered H3K79 trimethylation level (Fig. 4d), suggesting that the noncanonical enzyme activity of nuclear LDHA is required for H3K79 hypermethylation.

To test whether H3K79-specific methyl-transferase DOT1L was involved in α-HB-induced H3K79 hypermethylation, we adopted a DOT1L-specific inhibitor, EPZ004777[43], to delineate how LDHA-modulated H3K79 hypermethylation. EPZ004777 treatment blocked LDHA[NLS]-induced upregulation of H3K79 trimethylation (Fig. 4d and Supplementary Fig. 9c), indicating that DOT1L was necessary for nuclear LDHA to increase H3K79 methylation. To explore the mechanism underlying nuclear LDHA-induced H3K79 hypermethylation, coimmunoprecipitation was performed to determine the interaction between DOT1L and LDHA upon HPV16/18 E7 transduction or $H_2O_2$ treatment. Notably, both HPV16/18 E7 transduction and $H_2O_2$ treatment strengthened the binding of DOT1L with LDHA, which was partially blocked by NAC supplement (Fig. 4e, f, and

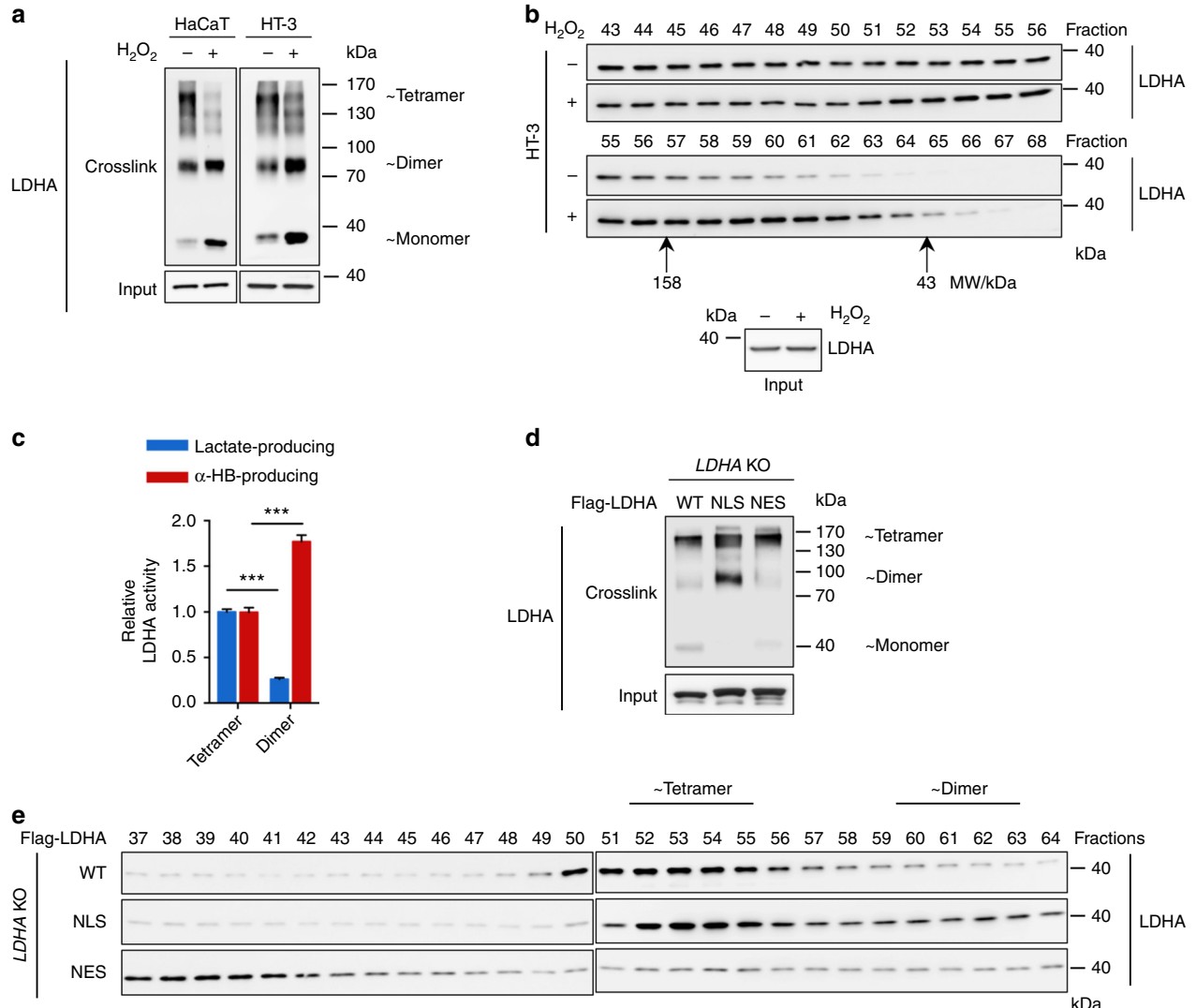

**Fig. 3** ROS disrupt LDHA tetramer formation and promote noncanonical enzyme activity. **a**, **b** ROS disrupt LDHA tetramer formation. HaCaT and HT-3 cell extracts with or without 10 μM $H_2O_2$ treatment were crosslinked by 0.025 % glutaraldehyde and analyzed by western blotting using LDHA antibody. Tetrameric, dimeric, and monomeric LDHA were indicated (**a**). HT-3 cell extracts were prepared from $1 \times 10^7$ cells with or without 10 μM $H_2O_2$ treatment and passed over the gel filtration column. Fractions were collected every 0.25 ml per tube and analyzed by western blot for LDHA protein. Molecular mass, 158 and 43 kDa marked below the blots, were determined by Gel Filtration Calibration Kit HMW (GE Healthcare). The loading inputs for gel filtration were shown below (**b**). **c** Dimer LDHA presents the noncanonical enzyme activity. The canonical and noncanonical LDHA enzyme activity assays were measured on tetramer fractions (Fraction #56, #57) and dimer fractions (Fraction #60, #61) separated from gel filtration. **d**, **e** More dimers form in the LDHA[NLS] group compare with that of LDHA[NES] group. Cell extracts from HEK293T *LDHA* KO cells expressing Flag-tagged WT, NLS, and NES LDHA were crosslinked by 0.025 % glutaraldehyde and analyzed by western blotting using LDHA antibody. Tetrameric, dimeric, and monomeric LDHA were indicated (**d**). Cell extracts from HEK293T *LDHA* KO cells expressing Flag-tagged WT, NLS, and NES LDHA were passed over the gel filtration column. Fractions were collected every 0.25 ml per tube and analyzed by western blot for LDHA protein. Molecular mass was determined by Gel Filtration Calibration Kit HMW (GE Healthcare). Results are representative of three independent experiments. All data are shown as mean ± SEM. The *p* values were determined by two-tailed *t*-test. The values of *p* < 0.05 were considered statistically significant. *, **, and *** denote *p* < 0.05, *p* < 0.01, and *p* < 0.001, respectively. NS means non significant

Supplementary Fig. 9d). Given that treatment of α-HB to LDHA[NES] cells restored H3K79 trimethylation, we hypothesized that α-HB produced by nuclear LDHA might take part in the activation of DOT1L though its binding to LDHA. To test this, we found that treatment of cells with α-HB alone increased the binding between endogenous DOT1L and LDHA (Fig. 4g). Overall, these results suggested that α-HB produced by nuclear LDHA induces H3K79 hypermethylation through promoting the interaction between DOT1L and LDHA, which possibly activates the methyl-transferase activity of DOT1L.

**NRF2 is required for LDHA-induced antioxidant responses.** High ROS levels are generally detrimental to cells, and the increased antioxidant capacity becomes vital to maintaining tumor development and cell survival. In addition, aberrant activation of Wnt/β-catenin signaling pathway, which is frequently observed in human cervical cancer, promotes cell proliferation and tumor progression[44,45]. Interestingly, the activation of DOT1L has been reported to modulate Wnt target genes expression[46,47]. To test whether the H3K79 hypermethylation induced by HPV16 E7 expression or $H_2O_2$ treatment further

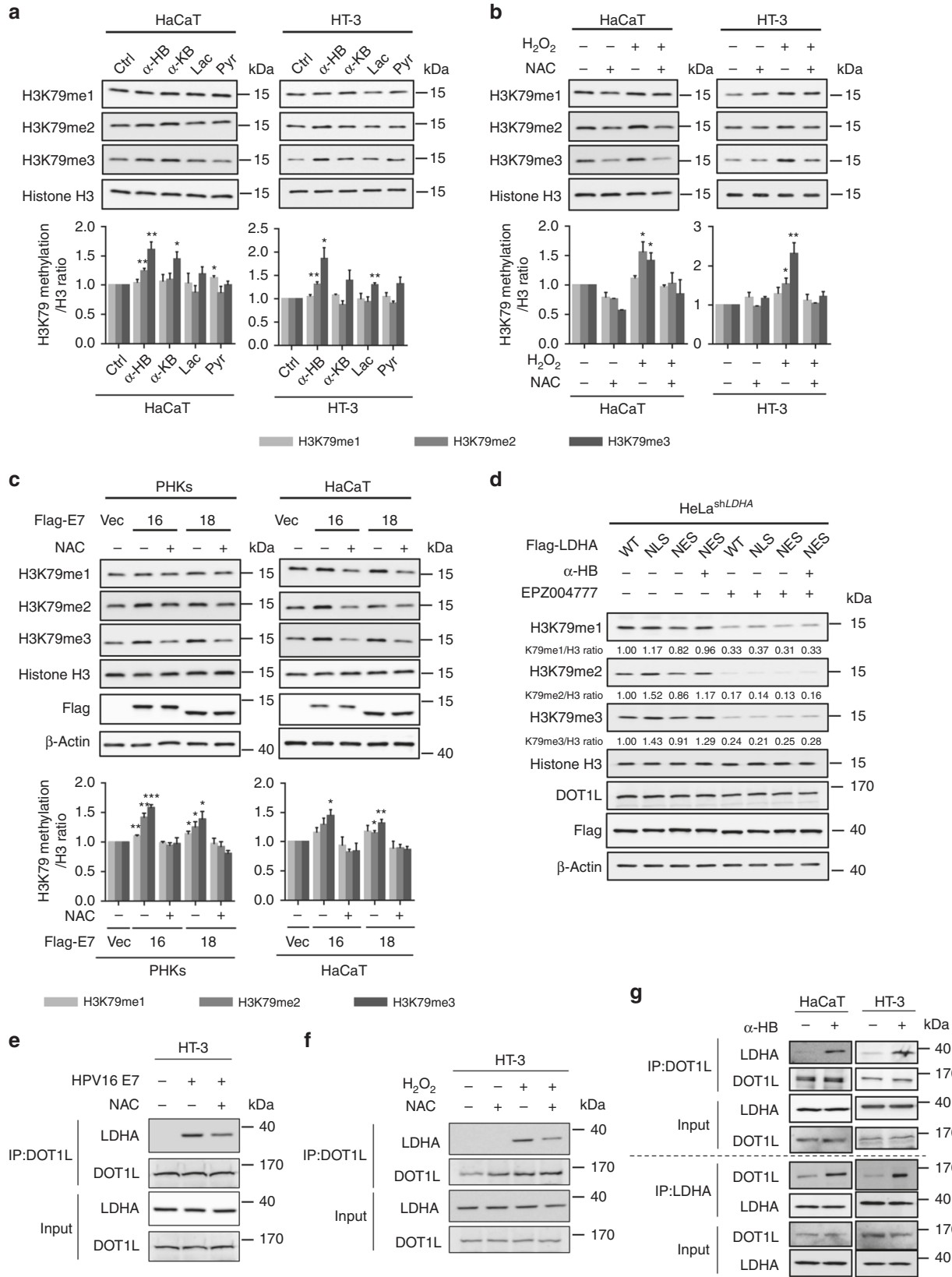

regulated cellular antioxidant responses or cell proliferation, seven antioxidant genes and three Wnt target genes were subjected to qPCR analysis. As expected, HPV16 E7 expression and $H_2O_2$ treatment increased the expression of the majority of these genes, which was blocked by NAC (Fig. 5a and Supplementary

Fig. 10a, b). Furthermore, ChIP-qPCR assays were carried out to define the histone methylation status of these HPV16/18 E7-induced target genes. We found that HPV16/18 E7 markedly enhanced H3K79 dimethylation level at the *SOD1*, *CAT*, *CTNNB1*, and *MYC* gene body, while NAC supplement reversed

**Fig. 4** DOT1L mediates α-HB-induced H3K79 hypermethylation. **a** α-HB upregulates H3K79 trimethylation. H3K79 methylation levels were analyzed in HaCaT and HT-3 cells upon 1 mM four LDHA-related metabolite treatments for 24 h. Ctrl, control; α-HB, sodium α-hydroxybutyrate; α-KB, sodium α-ketobutyrate; Lac, sodium l-lactate; Pyr, sodium pyruvate. H3K79 methylation / H3 ratio was quantified. **b** ROS increase H3K79 trimethylation. H3K79 methylation levels were analyzed in HaCaT and HT-3 cells upon 10 μM $H_2O_2$ for 6 h with or without 1 mM NAC pretreatment for 1 h as indicated. H3K79 methylation/H3 ratio was quantified. **c** E7 expression enhances H3K79 trimethylation. H3K79 methylation levels were analyzed in PHKs and HaCaT cells stably expressing vector or Flag-tagged HPV16/18 E7 coupled with or without 1 mM NAC treatment for 6 h. H3K79 methylation/H3 ratio was quantified. **d** DOT1L mediates nuclear LDHA-induced H3K79 hypermethylation. HeLa stable cells with *LDHA* knockdown and Flag-tagged Vec/WT/NLS/NES rescue were treated with or without 3 μM EPZ004777 for 48 h, combined with or without 1 mM sodium α-HB for 48 h as indicated. β-actin and histone H3 used as loading control in all immunoblots. **e** E7 triggers the binding of DOT1L with LDHA. Endogenous coimmunoprecipitation of DOT1L in HT-3 cells stably expressing vector or HPV16 E7 coupled with or without 1 mM NAC treatment for 6 h, and blotting of LDHA and DOT1L. **f** ROS trigger the binding of DOT1L with LDHA. Endogenous coimmunoprecipitation of DOT1L in HT-3 cells with or without 10 μM $H_2O_2$ for 6 h or 1 mM NAC for 6 h as indicated, followed by blotting with LDHA and DOT1L. **g** α-HB induces the binding of DOT1L with LDHA. Reversible endogenous coimmunoprecipitation of DOT1L and LDHA in HaCaT and HT-3 cells with or without 1 mM α-HB for 24 h, followed by blotting with LDHA and DOT1L. Results are representative of three independent experiments. All data are shown as mean ± SEM. The *p* values were determined by two-tailed *t*-test. The values of $p < 0.05$ were considered statistically significant. *, **, and *** denote $p < 0.05$, $p < 0.01$, and $p < 0.001$, respectively. NS means non significant

this effect (Fig. 5b and Supplementary Fig. 10c, d). These data suggest that HPV16/18 E7-induced H3K79 hypermethylation upregulates antioxidant genes and Wnt target genes expression in a ROS-dependent manner.

Several crucial transcription factors are enrolled in cellular redox sensing, such as NRF2, hypoxia-inducible factor 1α (HIF-1α), and nuclear factor kappa B (NF-κB)[19,48]. To further identify the key effector that directly mediated the antioxidant responses corresponding to high H3K79 methylation level, the expression of NRF2, HIF-1α, and NF-κB target genes were analyzed by qPCR. In response to HPV16 E7 expression and $H_2O_2$ treatment, the mRNA expression of NRF2 (*NQO1*, *GCLC*), HIF-1α (*SLC2A1*, *VEGFA*), and NF-κB (*IL-10*, *TNFB1*) target genes expression were remarkably increased (Fig. 5c and Supplementary Fig. 11a). However, EPZ004777 treatment blocked the upregulation of NRF2 target genes (*NQO1* and *GCLC*), but not HIF-1α or NF-κB target genes (Fig. 5c and Supplementary Fig. 11a). This data indicates that NRF2, not HIF-1α or NF-κB, may be the key transcription factor during HPV16 E7 or $H_2O_2$-induced H3K79 hypermethylation. Of note, we confirmed the increase of NRF2 protein level as a molecular marker of the NRF2 pathway activation upon HPV16/18 E7 expression (Supplementary Fig. 11b). We further validate the role of NRF2 in $H_2O_2$ or HPV16/18 E7-induced antioxidant genes expression. Increased mRNA expression of *NQO1* and *GCLC* was dramatically impaired by deletion of NRF2 or ML385, a NRF2-specific inhibitor[49] (Fig. 5d, e, and Supplementary Fig. 11c, d). To further test if LDHA is essential for the antioxidant gene activation, *LDHA* KO and *LDHA*, *NRF2* double-knockout (DKO) HeLa cells were generated for qPCR analysis. As shown in Fig. 5f and Supplementary Fig. 11e, *LDHA* KO significantly reduced *NQO1* and *GCLC* expression, but not the expression of HIF-1α or NF-κB target genes, upon $H_2O_2$ treatment. In addition, activation of antioxidant genes was markedly decreased in DKO cells (Supplementary Fig. 11f, g). Genetic ablation of *LDHA* remarkably blocked the increase of H3K79 dimethylation level of *SOD1*, *CAT*, *CTNNB1*, and *MYC*, while deletion of NRF2 blocked the increase of H3K79 dimethylation level of *SOD1* and *CAT*, but not *CTNNB1* and *MYC* gene body (Fig. 5g). Together, these data unveil a LDHA-DOT1L-NRF2 axis in HPV16/18 E7 and $H_2O_2$-induced antioxidant responses.

**Nuclear LDHA produced α-HB counteracts with oxidative stress**. Given the observed activation of antioxidant and Wnt signaling pathways, we presumed that nuclear LDHA enables cell survival and proliferation under oxidative stress. First, we examined the effect of nuclear LDHA on cell proliferation and colony formation in LDHA^WT, LDHA^NLS, and LDHA^NES cells,

respectively. There were no significant changes in cell growth between LDHA^WT, LDHA^NLS, and LDHA^NES stable cells under normal condition (Fig. 6a). However, LDHA^WT and LDHA^NLS stable cells demonstrated significant growth advantage and colony-formation ability under continuous oxidative stress compared with vector control and LDHA^NES cells (Fig. 6a–c and Supplementary Fig. 12a). Next, we examined the expression of antioxidant genes and Wnt target genes in LDHA^WT, LDHA^NLS, and LDHA^NES cells in response to $H_2O_2$. On one hand, LDHA^NLS cells presented higher expression levels of these genes under normal condition. On the other hand, LDHA^WT but not LDHA^NES cells showed more pronounced target genes expression in response to $H_2O_2$ (Fig. 6d and Supplementary Fig. 12b). Consistently, we observed that LDHA^NLS cells presented a lower level of ROS (Fig. 6e). Collectively, LDHA nuclear translocation is required for the activation of antioxidant genes and Wnt target genes.

As mentioned above, supplement of α-HB to LDHA^NES cells remarkably recovered H3K79 trimethylation level. We thus speculated that α-HB potentially rescued the antioxidant ability of LDHA^NES cells. Interestingly, the addition of α-HB remarkably relieved the ROS accumulation (Fig. 6f). In accordance, α-HB supplement significantly upregulated the expression of antioxidant genes and Wnt target genes (Fig. 6g). α-HB supplement effectively rescued cell growth and colony formation in LDHA^NES stable cells under continuous oxidative stress in a dose-dependent manner (Fig. 6h, i, and Supplementary Fig. 12c), suggesting that α-HB contributes to, at least in part, LDHA-mediated overcoming of oxidative stress. Collectively, these observations strongly support that LDHA nuclear translocation, in response to HPV induction or ROS stimulation, maintains redox homeostasis and cell survival under stressed conditions. LDHA gain-of-function that produces the antioxidant metabolite, α-HB, is critical for mediating antioxidant response.

**Nuclear LDHA implicates in cervical cancer development**. To examine the effect of nuclear LDHA in tumor growth, *LDHA* KO with vector, LDHA^WT, LDHA^NLS, and LDHA^NES rescue cells were injected subcutaneously into nude mice. Consistent with previous reports[32], LDHA^WT group showed increased tumor growth compared to vector group, while LDHA^NLS but not LDHA^NES group displayed significantly increased tumor growth compared with other groups (Fig. 7a–c). This result suggests that nuclear LDHA promotes tumor growth. In agreement, elevated NRF2, SOD1, MYC, and H3K79 trimethylation levels were observed in LDHA^NLS xenografts (Fig. 7d).

To examine the effect of HPV16-induced LDHA nuclear translocation in vivo, we adopted a K14-HPV16 transgenic mouse

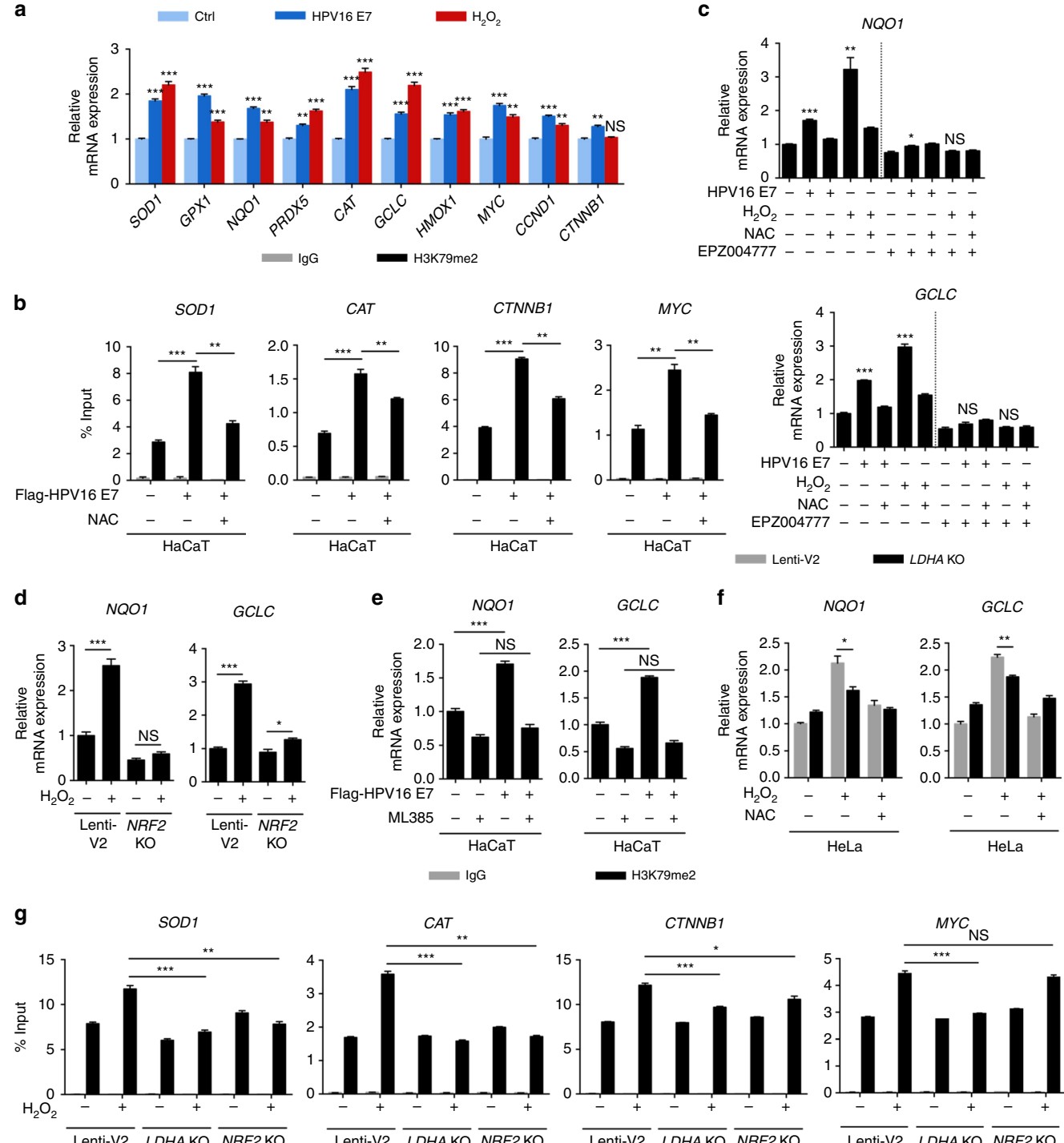

model with which direct cervical application of HPV16 E7-targeted transcription activator-like effector nucleases (TALENs) effectively mutated the *E7* oncogene, reduced viral DNA load, and restored retinoblastoma-associated protein (RB1) function[50]. Along with the TALEN applied, the cervical epithelium of the K14-HPV16 mouse showed a gradual loss of E7 and a reduction of epithelial proliferation. Meanwhile, the gradual loss of LDHA nuclear translocation correlated well with the decrease of H3K79 trimethylation level in serial sections (Fig. 7e). Strikingly, on day 24 when E7 expression was almost undetectable, nuclear LDHA was eliminated and H3K79 trimethylation levels were dramatically attenuated (Fig. 7e). These in vivo data strongly support that

HPV16 E7 induces LDHA nuclear translocation and H3K79 hypermethylation.

To further define the clinical relevance of our findings that HPV16 E7-induced LDHA nuclear translocation activated antioxidant response and Wnt signaling pathway, IHC analyses were performed to examine HPV16 E7 expression, LDHA nuclear localization, and H3K79 trimethylation levels in serial sections of 52 cases of human primary cervical cancer specimens. As shown in Fig. 7f and Supplementary Fig. 13, levels of HPV16 E7, nuclear LDHA expression, and H3K79 trimethylation were significantly positively correlated with each other, strongly supporting the crucial role of nuclear LDHA in HPV-positive cervical cancer.

**Fig. 5** NRF2 is required for nuclear LDHA-induced antioxidant responses. **a** E7 induces the expression of antioxidant and Wnt target genes. qPCR detecting antioxidant and Wnt target genes in vector (treated with or without $H_2O_2$) or HPV16 E7-expressing HT-3 cells treated with or without NAC as indicated. **b** E7 enhances the H3K79 dimethylation level at gene bodies. ChIP-qPCR showing the percentage of H3K79me2 enrichment at *SOD1*, *CAT*, *CTNNB1*, and *MYC* gene body relative to input genomic DNA in vector or HPV16 E7-expressing HaCaT cells treated with or without NAC. **c** EPZ004777 blocks increased *NQO1* and *GCLC* gene expression induced by E7 and $H_2O_2$. qPCR detecting *NQO1* and *GCLC* genes in vector (treated with or without $H_2O_2$) or HPV16 E7-expressing HaCaT cells treated with or without NAC and 3 μM EPZ004777 for 24 h as indicated. **d**, **e** NRF2 KO or NRF2 inhibition blocks increased *NQO1* and *GCLC* gene expression induced by E7 and $H_2O_2$. qPCR detecting *NQO1* and *GCLC* genes in HeLa *NRF2* KO cells treated with or without $H_2O_2$ (**e**), and vector or HPV16 E7-expressing HaCaT cells treated with or without 10 μM ML385 for 24 h (**f**) as indicated. **f** *LDHA* KO decreases *NQO1* and *GCLC* gene expression activated by $H_2O_2$. qPCR detecting *NQO1* and *GCLC* genes in HeLa *LDHA* KO cells upon $H_2O_2$ coupled with or without extended NAC treatment as indicated. **g** *LDHA* KO attenuates the H3K79 dimethylation level at gene bodies. ChIP-qPCR showing the percentage of H3K79me2 enrichment at *SOD1*, *CAT*, *CTNNB1*, and *MYC* gene body relative to input genomic DNA in HeLa *LDHA* KO and *NRF2* KO cells treated with or without $H_2O_2$. For ChIP-qPCR assay, rabbit IgG was included as a negative control. $H_2O_2$ was used as 10 μM for 6 h, and NAC was used as 1 mM for 6 h. Results are representative of three independent experiments. All data are shown as mean ± SEM. The *p* values were determined by two-tailed *t*-test. The values of $p < 0.05$ were considered statistically significant. *, **, and *** denote $p < 0.05$, $p < 0.01$, and $p < 0.001$, respectively. NS means non significant

## Discussion

LDHA is implicated in tumorigenesis and tumor development[32,51–53]. Loss of LDHA resulted in an elevated ROS production (Fig. 6e) and a concomitant decrease in cell proliferation and invasion[54,55]. Here, we have demonstrated a striking and previously unknown mechanism that nuclear LDHA senses ROS and gains a new catalytic function to promote HPV-induced cervical carcinogenesis: in response to HPV infection or oxidative stress, LDHA senses excessive ROS with a tetramer-to-dimer transition and nuclear translocation. Nuclear LDHA gains a noncanonical enzyme activity to produce an antioxidant metabolite, α-HB, which can protect cervical cancer cells from excessive oxidative stress and promote cell growth through epigenetic regulations. Interestingly, LDHA nuclear translocation appears to be essential for maintaining redox balance and sustaining cell proliferation.

LDHA's nuclear localization has been observed in multiple cancer types, including colorectal cancer, breast cancer, prostate cancer, lung cancer, and liver cancer[56] (reference to Human Protein Atlas available from www.proteinatlas.org), but the cause of LDHA nuclear translocation and the function of nuclear LDHA remains unclear. Our study discovered a ROS-dependent manner of LDHA nuclear translocation in response to HR-HPV infection, and α-HB produced by nuclear LDHA act as an important antioxidant metabolite facilitating cancer development. Given the high demand for ATP as "fuel" and metabolic intermediates as "building blocks" in cancer cells, aberrant proliferation generally accompanied with enhanced ROS production, suggesting that the mechanism of LDHA nuclear translocation is potentially ubiquitous. Our study provides a clue for how cancer cell balancing the demand of high proliferation rate and the high production of ROS: LDHA acts as a sensor for overloaded ROS, and then produce α-HB in the nucleus to enhance antioxidant capacity.

Nuclear pore complexes (NPCs) tightly control protein shuttling between cytoplasmic and nuclear. Protein is required to associate with importins or exportins to enter or exit the nucleus. Importins bind their cargo with NLS amino acid sequence in the cytoplasm, then interact with NPCs and pass through its channel. However, no classic NLS sequences were found in LDHA, suggesting the existence of noncanonical NLS signal or some unknown binding partners of LDHA facilitating its nuclear translocation.

Upon oxidative stress, an increased flux of cysteine into glutathione synthesis has been well-studied[57]. As a by-product of the methionine-to-glutathione pathway, α-HB production is directly linked to hepatic glutathione synthesis[58]. Our findings showed that α-HB level arises upon HPV16/18 E7 induction and cellular ROS accumulation in cervical cancer cells. Moreover, we

speculate that the accumulation of α-HB could further enhance antioxidant responses that may produce more α-HB, presenting a positive feed-forward loop. Similarly, β-hydroxybutyrate (β-HB) was found to suppress cellular oxidative stress through inhibition of histone deacetylase[59]. We hypothesized if LDHA exerted the enzyme activity to catalyze β-ketobutyrate (β-KB) to β-HB. However, no such enzyme activity was detected under our tested experiment conditions. Recent studies also discovered another noncanonical enzyme activity of LDHA, catalyzing α-ketoglutarate (α-KG) to α-hydroxyglutarate (α-HG), under acidic pH conditions[33,34]. However, the activity was not detectable under our tested experiment conditions.

We uncovered the α-HB accumulation increases H3K79 methylation levels, which is DOT1L-dependent. Furthermore, an interaction of DOT1L and LDHA has been identified upon α-HB treatment (Fig. 4g). However, how DOT1L methyl-transferase activity is regulated via its binding with LDHA remains unknown, which is worth further investigation.

In conclusion, we demonstrate that HPV16-induced nuclear translocation of LDHA is sufficient to trigger antioxidant responses and activate Wnt pathway, leading to cell survival and proliferation under oxidative stress (Fig. 7g). As such, blocking LDHA nuclear translocation may offer more opportunities to cervical cancer prevention and ROS-based cancer therapies.

## Methods

**Antibodies and reagents**. Antibodies against LDHA (Cell Signaling Technology, 3582, with 1:2500 working dilution for western blot and 1:500 working dilution for immunofluorescence; for mouse IHC, ABclonal, A1146, with 1:2000 working dilution), HPV16 E7 (Abcam, ab30731, with 1:2000 working dilution), Flag (Sigma, F3165, with 1:10000 working dilution for western blot and 1:1000 working dilution for immunofluorescence), Tubulin (Cell Signaling Technology, 3873, with 1:5000 working dilution), Lamin B1 (Cell Signaling Technology, 13435, with 1:2000 working dilution), H3K4me2 (Cell Signaling Technology, 9725, with 1:1000 working dilution), H3K9me2 (Cell Signaling Technology, 4658, with 1:1000 working dilution), H3K27me2 (Cell Signaling Technology, 9728, with 1:1000 working dilution), H3K36me2 (Cell Signaling Technology, 2901, with 1:1000 working dilution), H3K79me1 (Cell Signaling Technology, 12522, with 1:1000 working dilution), H3K79me2 (Cell Signaling Technology, 5427, with 1:1000 working dilution), H3K79me3 (Cell Signaling Technology, 4260, with 1:1000 working dilution; For IHC, Epigentek, A-4045, with 1:200 working dilution), H4K20me2 (Cell Signaling Technology, 9759, with 1:1000 working dilution), Histone H3 (Cell Signaling Technology, 4499, with 1:5000 working dilution), DOT1L (Abcam, ab72454, with 1:1000 working dilution), NRF2 (Abcam, ab62352, with 1:1000 working dilution), SOD1(Abcam, ab13498, with 1:2000 working dilution), MYC (Abcam, ab32072, with 1:2000 working dilution), β-actin (Invitrogen, MA5-15739, with 1:2000 working dilution) were purchased commercially.

Hydrogen peroxide solution ($H_2O_2$) (Sigma, 323381), N-acetyl cysteine (NAC) (Sigma, A7250), DAPI (Sigma, D9542), sodium pyruvate (Sigma, P5280), sodium (L) lactate solution (Sangon Biotech, A604046), sodium 2-ketobutyrate (Sigma, K0875), sodium 2-hydroxybutyrate (Santa Cruz, sc-258161), NADH (Sigma, N4505), EPZ004777 (Selleck, S7353), DPIC (Selleck, S8639), 50% glutaraldehyde solution (Sangon Biotech, A600875), and ML385 (MCE, HY-100523) were commercially obtained.

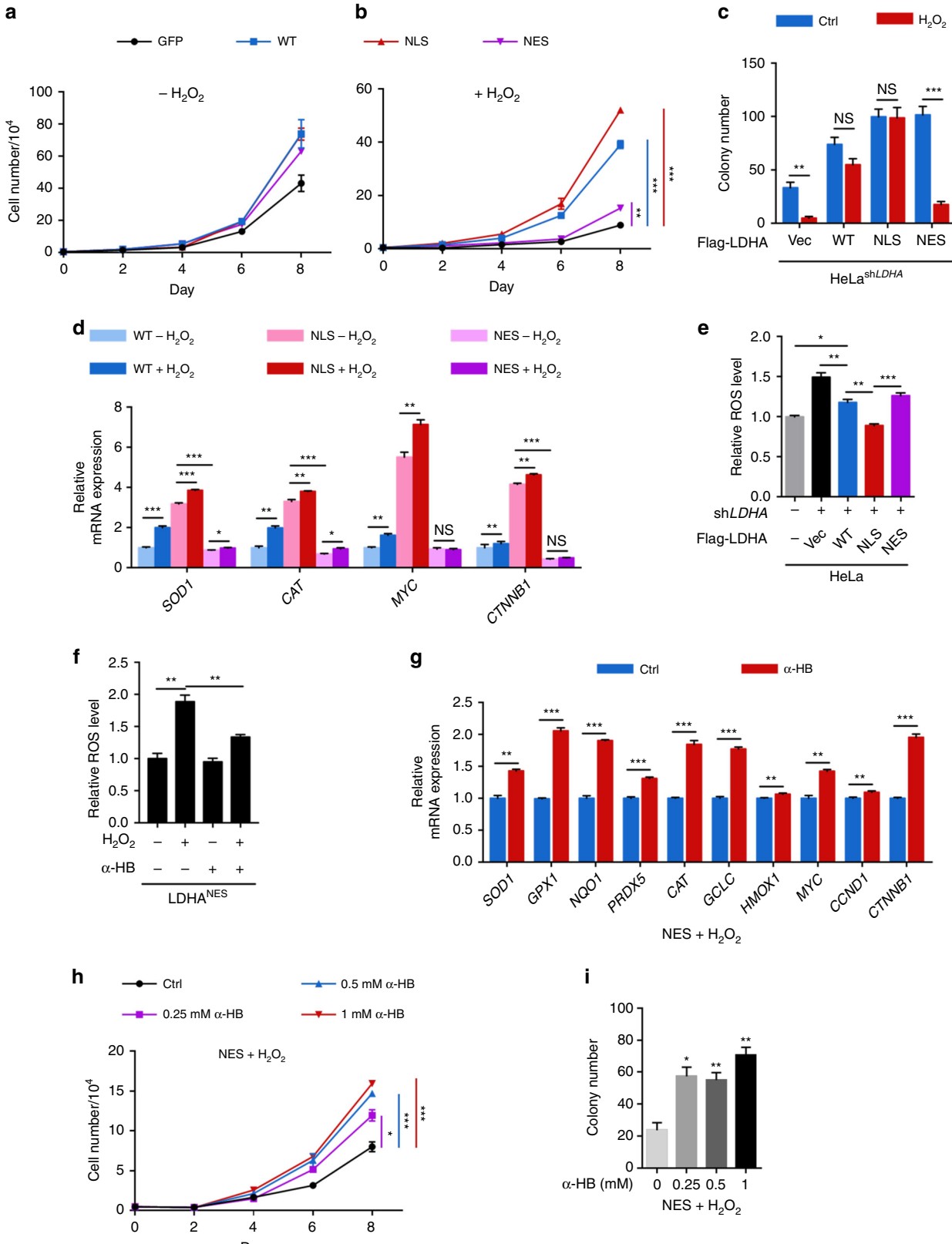

**Cell culture and treatment**. All the cell lines were purchased from American Type Culture Collection. HaCaT, HEK293T, SiHa, C33A, and HeLa cells were cultured in Dulbecco's Modified Eagle's Medium (DMEM) (Gibco), and HT-3 and U2OS cells were cultured in RPMI-1640 medium (Gibco) supplemented with 10% fetal bovine serum (Gibco) in the presence of penicillin, streptomycin (Gibco) at 37 °C in humidified atmosphere containing 5% $CO_2$.

Primary human cervix keratinocytes (PHKs) were isolated from normal cervical epithelial obtained from iCell (HUM-iCell-f016). PHKs were cultured in EpiLife Medium (Thermo Fisher, MEPI500CA) with the addition of EpiLife Defined Growth Supplement (EDGS) (Thermo Fisher, S0125). The Coating Matrix (Thermo Fisher, R011K) was used to enhance the attachment, growth, and population doubling potential of human keratinocytes. For all the experiments, keratinocytes cultured between the third and fifth passages were used.

**Fig. 6** Nuclear LDHA produced α-HB protects cervical cancer cells from oxidative stress. **a**, **b** Nuclear LDHA promotes cell growth under oxidative stress. Growth curve of HeLa stable cells with *LDHA* knockdown and Vec/WT/NLS/NES rescue were measured with or without 10 μM H$_2$O$_2$ treatment. **c** Nuclear LDHA increases colony formation under oxidative stress. Colony-formation assay of HeLa stable cells treated with or without 10 μM H$_2$O$_2$. **d** ROS enhance target gene expressions in LDHA$^{WT}$ and LDHA$^{NLS}$ but not LDHA$^{NES}$ stable cells. qPCR of antioxidant and Wnt target genes in HeLa stable cells with or without 10 μM H$_2$O$_2$ treatment for 6 h. **e** Nuclear LDHA downregulates cellular ROS level. Cellular ROS were measured in HeLa or HeLa stable cells using the ROS-sensitive fluorescent dye CM-H$_2$DCFDA by flow cytometry. **f** α-HB balances cellular ROS level in LDHA$^{NES}$ stable cells under oxidative stress. Cellular ROS were measured in HeLa LDHA$^{NES}$ stable cells treated with 1 mM sodium α-HB supplement for 48 h coupled with or without 50 μM H$_2$O$_2$ treatment for 30 min, using the ROS-sensitive fluorescent dye CM-H$_2$DCFDA by flow cytometry. **g** α-HB restores gene expressions in LDHA$^{NES}$ stable cells under oxidative stress. qPCR of antioxidant and Wnt target genes in HeLa LDHA$^{NES}$ stable cells treated with 10 μM H$_2$O$_2$ for 6 h, with or without 1 mM sodium α-HB supplement for 24 h. **h** α-HB rescues LDHA$^{NES}$ cell growth under oxidative stress. Growth curve of HeLa LDHA$^{NES}$ stable cells treated with 10 μM H$_2$O$_2$, with or without different dose of sodium α-hydroxybutyrate supplement. **i** α-HB restores LDHA$^{NES}$ colony formation under oxidative stress. Colony-formation assay of HeLa LDHA$^{NES}$ stable cells treated with 10 μM H$_2$O$_2$, with or without different dose of sodium α-hydroxybutyrate supplement. For cell proliferation and colony-formation assay, the media were exchanged every 24 h concerned for H$_2$O$_2$ decomposition. Results are representative of three independent experiments. All data are shown as mean ± SEM. The *p* values were determined by two-tailed *t*-test. The values of *p* < 0.05 were considered statistically significant. *, **, and *** denote *p* < 0.05, *p* < 0.01, and *p* < 0.001, respectively. NS means non significant

For H$_2$O$_2$ treatment and NAC supplement, cells were plated in complete culture media with 10 μM H$_2$O$_2$ for 6 h, then exchanged the medium supplemented with 1 mM NAC for another 6 h.

**Sample preparation, iodoTMT labeling, and mass spectrometry**. 5 × 10$^6$ cells cultured in media were washed in cold PBS, then lysed with lysis buffer (50 mM HEPES pH 7.9, 0.5 mM EDTA, 5 mM NaCl, 1% NP-40) adding 10 μl of LC/MS-grade methanol dissolved iodoTMT-126 reagent (Thermo Scientific, 90101) in an anoxic chamber. The lysates were sonicated for 30 s, then centrifuged at 16,000×g for 10 min at 4 °C to pellet insoluble materials. The supernatant was collected into a new tube and incubate at 900 rpm in ThermoMixer for 1 h at 37 °C protected from light. To remove the excessive iodoTMT-126 reagents, the proteins were precipitated with 500 μl of prechilled (−20 °C) acetone overnight at −20 °C. Precipitated proteins were washed twice with 500 μl of prechilled (−20 °C) acetone after centrifuged at 16,000×g for 30 min at 4 °C. The protein pellet was dissolved in 90 μl of denaturing alkylation buffer (6 M urea, 0.5% SDS, 10 mM EDTA, 200 mM Tris·HCl, pH 8.5) with 1 mM Tris (2-carboxyethyl) phosphine hydrochloride (TCEP), and adding 10 μl of LC/MS-grade methanol dissolved iodoTMT-127 reagent. The sample was incubated at 900 rpm in ThermoMixer for 1 h at 37 °C protected from light. Then, the iodoTMT-126/127-labeled protein was precipitated by prechilled (−20 °C) acetone as described.

For protein digestion, the precipitated proteins were dissolved in 100 μl 50 mM ammonium bicarbonate buffer, pH 8.0. Add 2.5 μl of trypsin to sample (1/50: trypsin/protein, w/w). Digest the sample overnight at 37 °C water bath. After digestion, peptides were cleaned up using C18 tips and lyophilized. For iodoTMT reagent enrichment, anti-TMT resin was used according to the manufacturer's protocol.

For LC-MS/MS analysis, purified peptides were resuspended in 1% acetonitrile and 1% formic acid (FA) in H$_2$O and analyzed by LC (Ultimate 3000, Thermo Scientific) equipped with self-packed capillary column (150 μm i.d. × 12 cm, 1.9 μm C18 reverse-phase fused-silica, Trap) using a 90 min nonlinear gradient at a flow rate of 600 nL/min. Eluted samples were analyzed by Q-Exactive HF (Thermo Fisher Scientific).

**Tissue DNA extraction and HPV detection**. The tissue DNA was extracted from 66 cervical cancer specimens using QIAamp DNA Mini Kit (Qiagen, 51304) following the manufacturer's protocol. The tissues were mechanically disrupted and lysed with proteinase K at 56 °C overnight, and DNA was purified using QIAamp Mini spin column. PCR-reverse dot blot (PCR-RDB) assay were performed using HPV Genotyping Detection Kit (Yaneng Biotech, Shenzhen, China) according to the manufacturer's instructions. The PCR reaction was amplified under the following conditions: an initial 50 °C for 15 min, 95 °C for 10 min; 40 cycles of 94 °C for 30 s, 42 °C for 90 s, and 72 °C for 30 s; and a final extension at 72 °C for 5 min. The PCR products were immobilized onto a nitrocellulose membrane and hybridized with typing probes. HPV subtypes were determined by the positive point on the HPV genotype profile on the membrane. HPV-positive and negative controls were also included in every experiment.

**Human cervical tumor samples and immunohistochemistry (IHC)**. Human cervical tumor samples were acquired from Sichuan Provincial People's Hospital and West China Second Hospital. Informed consents were obtained from the patients. The human studies were approved by the ethic committee of Sichuan Provincial People's Hospital and West China Second Hospital, Sichuan, China, and performed in compliance with the relevant ethical regulations. IHC staining was performed as previously described[60].

To quantify the expression of LDHA, HPV16 E7, and H3K79me3, five random views were microscopically examined and analyzed by experienced pathologists. Images were captured using a charge-coupled device camera and analyzed using Motic Images Advanced software (version 3.2, Motic China Group). ImagePro Plus (version 6.0, Media Cybernetics) was used for further quantification of DAB intensity of the image cubes. The DAB signal of the cytoplasmic and nuclear LDHA or the nuclear HPV16 E7 and H3K79me3 were measured in this study. The mean value of HPV16 E7 signal intensity of each case was categorized into the corresponding groups by the following scores: 0 (negative staining); 1 (weak staining); 2 (moderate staining); and 3 (strong staining). To quantify the distribution of the nuclear LDHA and H3K79me3 signal intensities on four HPV16 E7 intensity groups, the mean value of each case was categorized into score 0 to 12. Further analysis was based on the IHC scores of LDHA, HPV16 E7, and H3K79me3.

**Immunofluorescence and microscopy**. Cells were plated at 1 × 10$^6$ per well on glass coverslips in 6-well culture plates and stimulated as described above. Cells were fixed with 4% paraformaldehyde for 15 min at room temperature, then rinsed twice in PBS. Permeabilization was performed with 0.2% Triton-X-100 in PBS for 30 min at room temperature. Primary antibody incubation was performed overnight in blocking buffer (0.5% BSA in PBS) at 4 °C. The following day, cells were washed twice in PBS, then incubated 4 h in secondary antibody in PBS at 4 °C. DAPI in PBS (5 mg/ml) was added prior to imaging. Microscopy was performed with OLYMPUS IX81 system. Image analysis was performed with the program CellSens Standard.

**Quantification of cells with LDHA nuclear translocation**. Nuclear translocation of LDHA was calculated as the percentage of total cells in each individual immunostained image. Five images were quantified for each group. Five hundred or more cells were counted for each condition. White indicates the overlay of LDHA (green), Flag (red), and DAPI (blue) suggesting the nuclear translocation of LDHA in HPV16 E7-expressing cells. Cyan indicates the overlay of LDHA (green) and DAPI (blue) suggesting the nuclear translocation of LDHA. Representative immunostaining images of LDHA nuclear translocation were shown in Fig. 1d–g, and Supplementary Figs. 2c, 3b–d.

**Generation of stable cell pools**. To generate stable *LDHA* knockdown HeLa cell pools, shRNA targeting *LDHA* was constructed and targeting sequences were as follows:

sh*LDHA*-F: 5′-CCGGGCTACACATCCTGGGCTATTGCTCGAGCAATA GCCCAGGATGTGTAGCTTTTTG-3′

sh*LDHA*-R: 5′-AATTCAAAAAGCTACACATCCTGGGCTATTGCTCGAGC AATAGCCCAGGATGTGTAGC-3′

The retrovirus was produced by using a two-plasmid packaging system as previously described[61]. HeLa cells were infected with the retrovirus and selected with 2 μg/ml puromycin for 1 week. To generate *LDHA* knockdown and rescue stable cell pools, Flag-tagged human wild-type (WT), nuclear localization signal (NLS), or nuclear export signal (NES) LDHA was cloned into the retroviral pQCHIX vector and cotransfected with vectors expressing the gag and vsvg genes in HEK293T cells to produce retroviruses, HeLa cells with stable *LDHA* knockdown were then infected, following a selection with 200 μg/ml hygromycin B for 1 week. The NLS and NES sequences were listed as follow:

NLS: 5′-CCTAAGAAGAAGAGGAAGGTT-3′
NES: 5′-CTTCAGCTACCACCGCTTGAGAGACTTACTCTT-3′.

**Generation of knockout cells**. CRISPR-Cas9-based gene knockout was performed by oligonucleotides, three sgRNAs containing the *LDHA*-targeting sequences, and two sgRNAs containing *NRF2*-targeting sequences using lentiCRISPRv2 (Addgene) system. The targeting sequences were as follow:

*LDHA* sgRNA-1: 5′-CATTAAGATACTGATGGCAC-3′
*LDHA* sgRNA-2: 5′-CCGATTCCGTTACCTAATGG-3′

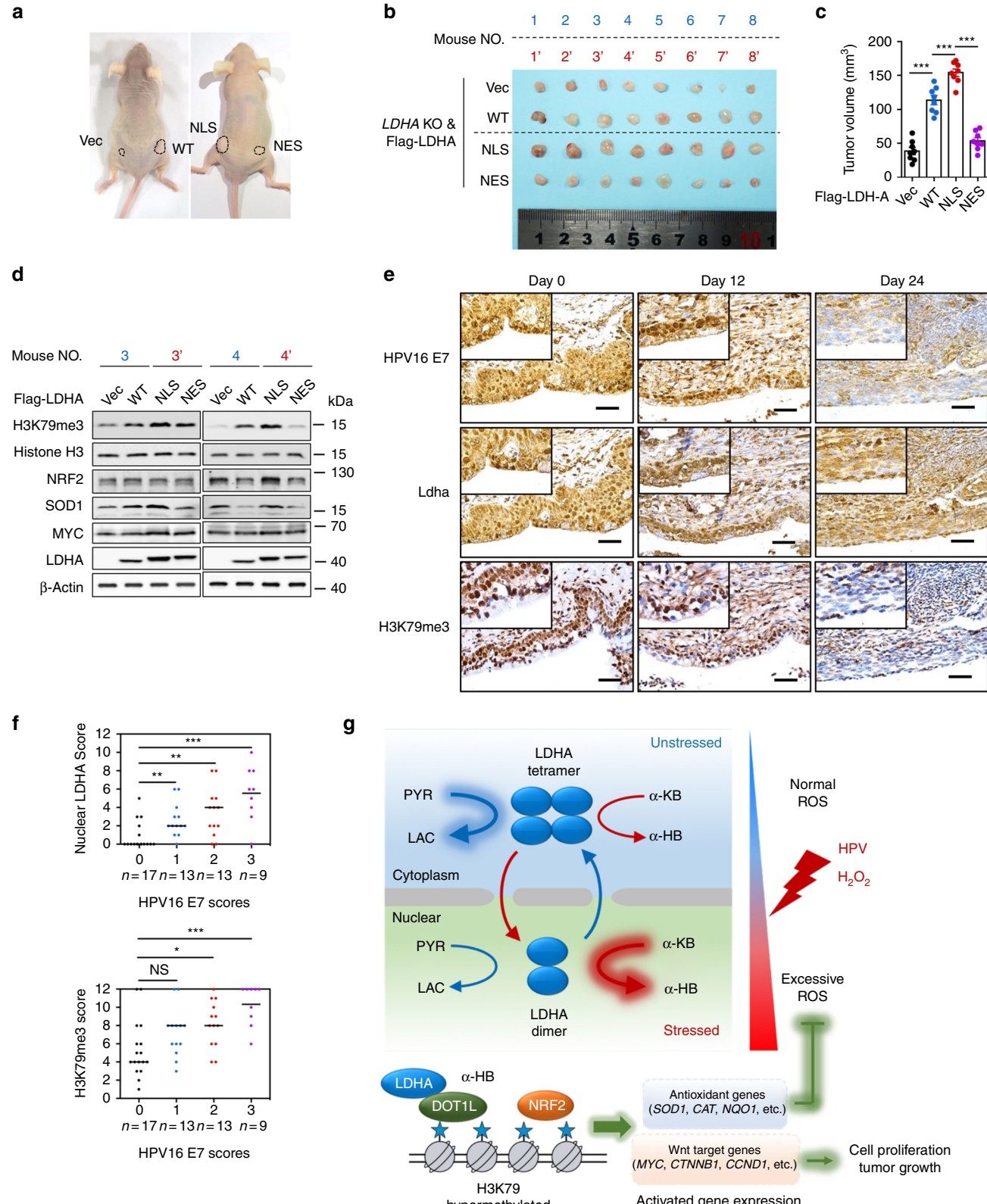

LDHA sgRNA-3: 5′-CCCGATTCCGTTACCTAATG-3′
NRF2 sgRNA-1: 5′-CCCGTCCCGGCACCACCGCA-3′
NRF2 sgRNA-2: 5′-TGGGACGGGAGTCCCGGCGG-3′.

**Measurement of intracellular and mitochondrial ROS levels**. ROS and mitochondrial superoxide production were determined using a fluorescent dye chloromethyl-2′,7′-dichlorofluorescein diacetate (H$_2$DCF-DA, Sigma, 35845) and a specific mitochondrial superoxide indicator MitoSOX Red (Thermo Fisher, M36008), respectively. Briefly, cells with specified treatments were washed with PBS and incubated with 5 μM H$_2$DCF-DA (or 10 μM MitoSOX) at 37 °C for 30 min to load the fluorescent dye. Afterward, cells were washed twice with PBS, and trypsinized, centrifuged, resuspended in 500 μl PBS. Cells were kept in the dark until analysis by flow cytometry (Accuri C6, BD Biosciences).

**Fig. 7** HPV16 E7-induced LDHA nuclear translocation implicates in cervical cancer development. **a**, **b**, **c** LDHA$^{NLS}$ increases tumor growth in xenograft nude mice model. Representative images (mouse NO. 1 with Vec on the left flank and WT on the right flank, and mouse NO. 1′ with NLS on the left flank and NES on the right flank) of HeLa *LDHA* KO with Vec/WT/NLS/NES putback xenograft mice (**a**). Dissected tumors in xenograft mice transplanted with HeLa *LDHA* KO with Vec/WT/NLS/NES putback cells (**b**) and the tumor volumes on day 30 were represented (**c**). **d** LDHA$^{NLS}$ tumor shows an elevated H3K79me2, NRF2, SOD1, and MYC levels. The level of H3K79me2 and the expression of NRF2, SOD1, and MYC in two pairs of representative tumors were analyzed by western blotting. β-actin and histone H3 were used as loading control. **e** LDHA nuclear translocation are positively correlated with H3K79 trimethylation in K14-HPV16 transgenic mice. Representative IHC images of HPV16 E7 expression, LDHA localization, and H3K79me3 level at indicated time point in TALEN-mediated targeting E7 of K14-HPV16 transgenic mice. *n* = 3 per group. Scale bar, 50 μm. **f** HPV16 E7 levels, LDHA nuclear translocation and H3K79 trimethylation are positively correlated with each other in human cervical tumor tissues. The 52 cases were divided into four groups on the basis of their tumor HPV16 E7 expression scores. Horizontal lines represent the median. Data are shown as mean ± SEM. The *p* values were determined by two-tailed *t*-test. The values of *p* < 0.05 were considered statistically significant. *, **, and *** denote *p* < 0.05, *p* < 0.01, and *p* < 0.001, respectively. NS means non significant. **g** Proposed molecular mechanism model of a noncanonical role for LDHA in responds to HR-HPV infection. PYR, pyruvate, LAC, lactate, α-KB, α-ketobutyrate, and α-HB, α-hydroxybutyrate

**Nuclear isolation**. Isolation of nuclei was performed using the commercially available nuclei isolation kit: nuclei PURE prep from Sigma Aldrich (NUC201). Briefly, adherent cells were washed with PBS and scraped from the plate in the presence of lysis buffer. Cells (in lysis media) were carefully placed on top of a 1.8 M sucrose gradient and the resulting suspension was centrifuged at 30,000×g for 45 min in a precooled swinging bucket ultracentrifuge. Nuclei were collected as a white pellet at the bottom of the centrifuge and washed with nuclei storage buffer (provided within the kit). Isolated nuclei were immediately subjected to metabolite quantification.

**LDHA enzyme activity assay**. The canonical and noncanonical LDHA enzyme activity was determined as described previously[62]. For endogenous, cells with specified treatments were lysed by NP-40 buffer [50 mM Tris-HCl (pH 7.4), 150 mM NaCl, 0.3% Nonidet P-40] containing protease inhibitors [1 mg/ml aprotinin, 1 mg/ml leupeptin, 1 mg/ml pepstatin, 1 mM Na$_3$VO$_4$, and 1 mM phenylmethylsulfonyl fluoride (PMSF)], and for exogenous, Flag-tagged WT/NLS/NES LDHA proteins were overexpressed in cells, immunoprecipitated with Flag-beads, eluted by Flag peptides (Gilson Biochemical), and subjected into activity assay with pyruvate and NADH as substrates for canonical enzyme activity and α-ketobutyrate (α-KB) and NADH as substrates for noncanonical enzyme activity, respectively. Reaction mixture consists of 50 mM Tris-HCl (pH 7.4), 20 μM NADH, 2 mM pyruvate (or 3.3 mM α-KB) in a total volume of 500 μL. Reactions were initiated by adding the enzyme and analyzed at 25 °C. Activities were measured by the conversion of NADH to NAD$^+$, which was monitored by measuring the decrease of fluorescence (Ex. 350 nm, Em. 470 nm, HITACH F-4600 fluorescence spectrophotometer) for NADH decomposition. LDHA enzyme activities were normalized to LDHA protein level.

**Metabolite extraction**. In total, $2 \times 10^6$ cells or nuclei were resuspended in 800 μl of ice-cold 80% methanol and 20% ddH$_2$O. Samples were vigorously vortexed and placed in liquid N$_2$ for 10 min to freeze. Then thawed on ice for 10 min, before the freeze-thaw cycle was repeated. Samples were centrifuged at 13,000 rpm to pellet cell debris, lipids and proteins. The supernatant was evaporated and resulting metabolites were resuspended in HPLC-grade H$_2$O. Metabolites were normalized to protein concentration.

**Metabolite analysis by LC-MS/MS**. The metabolite samples were resuspended in 20 μl HPLC-grade H$_2$O for mass spectrometry. Three microliter of sample extract was injected and analyzed using a 6500 Q-Trap triple quadrupole mass spectrometry (AB Sciex) coupled to a Prominence HPLC System (Shimadzu) via multiple reaction monitoring (MRM). Metabolites were targeted in negative ion mode (IonSpray Voltage: −4500V). Ultimate AQ-C18 column (1.7 μm, 2.1 × 250 mm, Welch) was used for separation of pyruvate, lactate, α-KB, and α-HB. Two gradient programs were applied: (1) for pyruvate and lactate. Mobile phase A: 0.01% formic acid in water; mobile phase B: acetonitrile-methanol (1:1); flow rate: 0.400 mL/min; gradient: starting from 0% phase B for 1.0 min, increased to 90% B from 1.0 to 4.0 min, 90% B was held from 4.0 to 5.0 min, 90% B to 0% B from 5.0 to 6.0 min, 0% B was held for 4.0 min to re-equilibrate the column. The stop time was 10.0 min. (2) For α-KB and α-HB. The mobile phase A, B, and flow rate were the same as 1). Gradients were run starting from 0% phase B for 1.0 min, increased to 60% B from 1.0 to 10.0 min, 60% B to 80% B from 10.0 to 11.0 min, 80% B was held from 11.0 to 13.0 min, 80% B to 0% B from 13.0 to 13.1 min, 0% B was held for 6.9 min to re-equilibrate the column. The stop time was 20.0 min. The other parameters were kept constant as follows: curtain gas (CUR) was kept at 35 psi, ion source gas 1(GS1) was kept at 40 psi, ion source gas 2 (GS2) was kept at 40 psi, and the drying temperature was maintained at 600 °C. Monitored transitions: pyruvate: m/z 87.0 > 43.0 [collision energy (CE): 10 V], lactate: m/z 89.0 > 43.0 (CE: 13 V), α-KB: m/z 101.0 > 57.0 (CE: 10 V), α-HB: m/z 103.0 > 57.0 (CE: 13 V). Peak areas from the total ion current for each metabolite were integrated using Skyline software (AB Sciex).

**α-HB standard curve construction**. The standard curve was prepared with seven concentrations of sodium α-hydroxybutyrate. The calibrators for sodium α-hydroxybutyrate was prepared at the following concentrations: 0.1, 0.5, 1.0, 5.0, 10.0, 25.0, 50.0 μM. The standard curve was determined using the linear regression of the peak area of sodium α-hydroxybutyrate against the interval standards. Quantification of the metabolite samples was calculated using interpolation.

**Protein crosslinking assay**. HaCaT and HT-3 cells treated with or without 10 μM H$_2$O$_2$ for 6 h and HEK293T *LDHA* KO cells expressing Flag-tagged WT, NLS, and NES LDHA were lysed by HEPES lysis buffer (40 mM HEPES pH 7.5, 150 mM NaCl, 0.1% NP-40) containing protease inhibitors for 30 min at 4 °C. Cell lysates were clarified by centrifuge at 16,000×g for 30 min at 4 °C. For protein crosslinking reactions, the supernatants were added with 0.025 wt. % glutaraldehyde for 30 min at 37 °C. The reactions were terminated by a final concentration of 50 mM Glycine. Samples were then separated by 8% SDS-PAGE and analyzed by western blotting with LDHA antibody.

**Gel filtration**. Cells were treated with or without 10 μM H$_2$O$_2$ for 6 h and lysed by NP-40 buffer containing protease inhibitors (as above) for 30 min and centrifuged for 15 min at 13,000 rpm to remove cell debris. The gel filtration column (Superdex 200 Increase; GE Healthcare) was washed and equilibrated by cold PBS. Extracts were passed over the gel filtration column. The speed rate of flow is 0.4 ml/min. Fractions were collected every 0.25 ml per tube and analyzed by western blotting. Molecular mass was determined by Gel Filtration Calibration Kit HMW (GE Healthcare).

**Coimmunoprecipation and western blotting**. Cells were lysed in ice-cold NP-40 buffer containing protease inhibitor (as above). The whole-cell lysates were incubated with protein A/G Sepharose beads (Roche) for 2 h at 4 °C and the beads were discarded to eliminate the non-specific binding. Then, the supernatants were incubated overnight with DOT1L antibody at 4 °C, followed by incubating with protein A/G Sepharose beads for another 2 h at 4 °C. Afterward, western blotting with indicated antibodies was performed. Standard western blotting protocols were adopted. Relative protein levels were quantified by measuring the band intensity of western blots by using Image J software. All the uncropped, full-size scans of western blots are presented in Supplementary Fig. 14.

**Quantitative RT-PCR**. Total RNA was isolated from cultured cells using Trizol reagent and reverse transcribed with random primers following the manufacturer's instructions (TaKaRa). The cDNA was preceded to real-time PCR with gene-specific primers in the presence of SYBR Premix ExTaq (TaKaRa). PCR reactions were performed in triplicate on Roche LightCycler 480 System and the relative amount of cDNA was calculated by the comparative CT method using the β-actin as a control. Primer sequences were provided in Supplementary Data 3.

**Chromatin immunoprecipitation (ChIP)-qPCR assays**. ChIP-qPCR assays were performed as previously described[60]. Briefly, $2 \times 10^6$ HaCaT, HT-3, and HeLa cells were crosslinked with 1% paraformaldehyde, lysed and sonicated using the Bioruptor at high-output power setting for 40 cycles (30 s ON and 30 s OFF). Solubilized chromatin was immunoprecipitated with ChIP-grade antibody for H3K79me2 or rabbit IgG (negative control), following that the antibodies were preincubated with protein A sepharose beads overnight at 4 °C. Antibody-chromatin complexes were pulled down by protein A sepharose beads (Santa Cruz), washed with high salt buffer, LiCl buffer, TE buffer, and then eluted. After crosslink reversal in a water bath at 65 °C for 4 h and proteinase K digestion for 1 h at 55 °C, the ChIPed DNA was purified with QIAquick PCR Purification Kit (Qiagen, 28106). The DNA fragments were detected by qPCR. Histone H3K79 dimethylation marks were mapped at particular gene body regions of target genes (*SOD1*, *CAT*, *CTNNB1*, and *MYC*). Primers for the regions with peaks were

designed for ChIP-qPCR analysis. All tested primers targeting *SOD1*, *CAT*, *CTNNB1*, and *MYC* are listed in Supplementary Data 3.

**Cell proliferation assay and colony-formation assay**. Cells were trypsinized, resuspended in PBS, and counted by using a hemocytometer. For cell proliferation assay, cells were plated in 6-well plates at 2500 cells per well in 2 ml media with or without $H_2O_2$, and plates were counted at 2, 4, 6, and 8 days. For colony-formation assay, cells were seeded in 6-well plates at 500 cells per well in 2 ml media with or without 10 μM $H_2O_2$. After 14 days, colonies were fixed in 4% paraformaldehyde and stained with 0.2% crystal violet. Colonies with >50 cells were counted. The media were exchanged every 24 h concerned about $H_2O_2$ decomposition.

**Xenograft model**. The procedures related to animal subjects were approved by ethic committee of the Institutes of Biomedical Sciences (IBS), Fudan University, Shanghai, China. In total, $5 \times 10^6$ HeLa *LDHA* KO cells with Vec/WT/NLS/NES LDHA put back were subcutaneously injected into the flanks of athymic nude mice (6-week-old males): Vec on the left flank and WT on the right flank for mouse NO. 1–8, while NLS at the left flank and NES at the right flank for mouse NO. 1′–8′. Tumor volume was recorded by caliper measurements using the formula (length [mm]) × (width [mm]) × (height [mm]) × (π/6). Tumors were dissected and weighed at day 30, and tumor volumes derived from *LDHA* KO cells with Vec/WT/ NLS/NES LDHA putting back were compared. All the animal studies were performed in compliance with the relevant ethical regulations.

**TALEN targeting HPV16 E7 in K14-HPV16 transgenic mice**. K14-HPV16 transgenic mice have been described previously[63]. Breeding pairs of K14-HPV16 transgenic mice were provided by the National Cancer Institute (NCI) Mouse Repository (Frederick, MD, USA) [strain nomenclature: FVB.Cg-Tg (KRT14-HPV16) wt1Dh] and bred at the Experimental Animal Center, HUST. HPV16-positive female mice (6 to 8 weeks old) were randomly grouped. The DNA of TALEN plasmid was mixed with TurboFect In Vivo Transfection Reagent (R0541, Fermentas, Thermo Scientific) according to the manufacturer's protocol. Mice were anesthetized with 3% pentobarbital sodium (intraperitoneal injection of 50 mg/kg mice body weight), and the polymer–DNA complexes were pipetted into the vagina using 200 μl pipet tips after washing with saline (in a maximum volume of 20 μl)[64]. Then, mice were kept in the dorsal position for at least 40 min under anesthesia to avoid loss of the complexes. The mice were euthanized after pentobarbital anesthetization at the indicated times. The vagina was dissected, fixed in 4% paraformaldehyde, and sectioned for IHC. The protocol for the assembly of the TALEN targeting HPV16 E7 was described previously[50]. The TALEN target sequence was as follows: 5′-ATGTTAGATTTGCAACCAGAGACAACTgatctctactgttatgagc AATTAAATGACAGCTCAGAGGAGG-3′. Uppercase letters: left and right target sequences of TALEN, lowercase letters: spacer sequences.

**Statistical analysis**. Statistical analyses were performed with a two-tailed unpaired Student's *t*-test. All data shown represent the results obtained from three (or as indicated) independent experiments with standard errors of the mean (mean ± SEM). The values of $p < 0.05$ were considered statistically significant. *, **, and *** denote $p < 0.05$, $p < 0.01$, and $p < 0.001$, respectively. NS means non significant.

## Data availability

The mass spectrometry proteomics data has been deposited to the ProteomeXchange Consortium via the PRoteomics IDEntifications (PRIDE) partner repository[65] with the dataset identifier PXD011058. The datasets generated and analyzed during the current study are available within the article and the Supplementary Information files, and from the corresponding author upon reasonable request.

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

## Acknowledgements

We thank the members of Fudan Cancer Metabolism laboratory for discussion throughout this study. We also thank the Biomedical Core Facility of Fudan University for their technical support. This work was supported by MOST (Grant No. 2015CB910401), NSFC (Grant No. 81790253, 81790251, 81821002, 81430057, 81430071, and 81672381) to Q.Y.L and C.H, Sichuan Science and Technology Program (Grant No. 2018RZ0133) to C.H, and Shanghai Municipal Education Commission (Grant No. N173606) to Q.Y.L.

## Author contributions

Y.L. and J.Z.G. performed the experiments, analyzed the data and co-wrote the paper. Y.L. designed and performed histopathological experiments and analysis. K.W. aided in paper preparation. W.D. and H.W. conducted K14-HPV16 transgenic mouse trials. X.L., S.Z. and L.Z. collected human samples and informed consents. X.C.L. performed the protein crosslink experiments. H.B.Y. performed the enzymatic experiments and analyzed the data. C.X. assisted histopathological experiments and analysis. W.G. assisted schematic graph design and drawing. Y.P.W. designed a part of the study and helped to interpret part data, W.H. helped to interpret part data, Y.W. supported the research. C.H. and Q.Y.L. conceived the idea, designed and supervised the study, analyzed the data, and co-wrote the paper

## Additional information

**Competing interests:** The authors declare no competing interests.

