## [Peer Review File · Nature Communications]

Reviewers' Comments:

Reviewer #1:

Remarks to the Author:

In this manuscript, the authors provide evidence that ROS induction or expression of the oncoprotein E7 from the mucosal high (HR) human papillomavirus type 16 (HPV16) induces nuclear translocation of the dimeric form of lactate dehydrogenase A (LDH-A). This nuclear form of LDH-A promotes the accumulation of α -hydroxybutyrate (α -HB) via a non-canonical activity. High nuclear levels of α -HB lead to histone H3 Lys79 hypermethylation, resulting in the activation of genes encoding proteins involved in the antioxidant responses and Wnt signalling pathway. Overall, the experiments are well performed. The authors' conclusions concerning the link between HPV carcinogenesis and LDH-A are less convincing and require further work and clarifications as described in detail below. In addition, no experiments were performed in primary human keratinocytes that are naturally infected by the different HPV types.

Specific points

(1) Results, lines 48-50: "we collected HPV-positive (n = 38) and HPV-negative (n = 28) cervical cancer specimens from 66 patients". It is well known that the majority of cervical cancers are positive for HPV16 or other 11 HR HPV types, e.g. 18, 31, 33, 35 45, etc. Therefore, it is unlikely that the authors managed to identify 28 HPV-negative cervical cancers. No information is provided about the characterization of the 66 cervical cancers. Did the authors check for the presence of DNA and RNA of the different HR HPV types? This analysis is essential for the classification of HPV-negative or HPV-positive cervical cancers. Without this additional work, the interpretation of the results shown in Figure 1 and 6 is impossible.

(2) Figure 1c: More fields of the IF staining should be shown to better demonstrate the LDH-A nuclear translocation in HPV16 E7 HT-1 cells.

(3) Figure 3c: HeLa cells express E6 and E7 from HR HPV type 18. Why is LDH-A not in the nucleus of these cells? Is the ability to promote LDH-A translocation into the nucleus a feature of HPV16, and not of HPV18 or other HR HPV types? Since the authors state throughout the manuscript that they have identified a key event in HPV-mediated carcinogenesis, a comparative analysis of the properties of the different HR HPV oncoproteins in promoting LDH-A nuclear accumulation should be performed. Ideally, primary human keratinocytes should be used as an experimental model, since they represent the natural target of HPVs. Regarding the experiments performed in HeLa cells, without the demonstration that HPV18 E7 is not able to promote LDH-A nuclear translocation, it is difficult to interpret the results obtained in HeLa cells shown in many Figures of the manuscript.

(4) Figure 3b: gel filtration fractionations using cytoplasmic and nuclear extracts should be added to corroborate their interpretation of the results.

(5) Figures 4a and b: quantification of the protein band in at least three independent experiments should be also shown.

(6) Figure 4e: The mechanism of the interaction LDH-A and DOT1L should be better investigated. Is this interaction mediated by post-translation modifications of these proteins?

Minor points

(1) Abstract, lines 4-5: "High-risk human papilloma virus (HPV) infection is the main cause of cervical cancer, however the underlying mechanism remains poorly defined". This statement is incorrect. In the last 20 years, an enormous number of studies have elucidated many mechanisms of E6 and E7 oncoproteins involved in cellular transformation. The authors could check some of the

most recent reviews on HPV and carcinogenesis.

(2) Abstract: please give the definition of DOT1-like (Disruptor of telomeric silencing 1-like).

(3) Introduction: line 29 "...but the underlying mechanism remains undefined." Please see point 1 above.

(4) Introduction, lines 24-26: "During high-risk HPV infection, viral differentiation-dependent promoters become upregulated that is characterized by the increased expression of two viral early genes, E6 and E7". As far as I know, only the activity of the late promoter, which controls L1 and L2 expression, is regulated in a differentiation-dependent manner. The accumulation of E6 and E7 in the suprabasal layers of the infected epithelium is mainly explained by a high HPV genome amplification.

(5) Legends Figure 3b and d should provide a short explanation for the numbers "158" and "43", which most likely represent the molecular weight markers.

(6) Line 128: The sentence refers to Figure 3e and not 3d.

(7) Reference 8 is not complete

Reviewer #2:

Remarks to the Author:

Nuclear LDH-A Produces α -Hydroxybutyrate to Activate Antioxidant Responses and Promote HPV-induced Cervical Carcinoma

The authors present evidence supporting a model in which infection of epithelial cells with high risk HPV results in a burst of reactive oxygen species, translocation of LDH to the nucleus and activation of a gene profile that supports the growth of cervical cancer. The experiments appear to be well done as far as they go, but the overall model has some significant gaps in logic and experimental support. First of all, the introduction of HPV E7 into untransformed cervical cells is not equivalent to infection with replication competent virus. Therefore, data is necessary to support the conclusion that E7 alone can induce ROS. How does this occur? Is there altered mitochondrial function? P450 activation? And if this is such a stressful event, why does it not activate the more conventional KEAP-NRF pathway to detoxify the species, rather than invoking a whole new molecular mechanism?

If the reader accepts that LDH does translocate to the nucleus after E7 expression, then the easy explanation of its effect there would incorporate some level of redox signaling involving altered NAD⁺/NADH ratios. As an enzyme that typically consumes NADH while converting pyruvate to lactate, this would significantly decrease antioxidant pools in the nucleus, allowing for increased oxidative stress. This would seem much more likely to alter gene expression (as several transcription factors have been shown to be redox sensitive (for review Cellular Signaling Volume 14, Issue 11, November 2002, Pages 879-897). The use of NAC as proof of mechanism is not really that conclusive because NAC will influence all redox balances in the cell (even nuclear levels of NAD⁺/NADH). Additionally, the effects reported on H3K79 hypermethylation need to be more closely tied to changes in the expression of the genes listed in figure 4f.

Finally, can the authors show any significant effect of nuclear LDH on characteristics of cellular transformation? Does the expression of LDHnls versus LDHnes versus LDHwt alter growth of tumor cells in nude mice? Can any increase be directly attributed to gene expression changes identified here?

Major points.

1. In figure 1f, the other cellular fractions need to be run and compared to the nuclear fraction. How much of the LDH is nuclear? How much of the LDH translocates to the nucleus after E7 expression?
2. What are the relative lactate producing versus aHB producing activities of LDH? All the reported activities appear to be normalized to sit is difficult to tell. Is the aHB producing activity half of the lactate producing activity? 1%? How much extra aHB is produced by the increase the aHB activity of only 1.5 fold?
3. The biochemical fractionation of dimeric versus tetrameric LDH is totally unconvincing. Why did then authors only show fractions 43-68? Does fraction 43 represent the flow through? If the monomer is 36kDa, then the dimer is 70 kDa. There appears to be some LDH in this region between 43 and 168 in the H₂O₂ treated cells, but this looks to be a very small fraction of the total LDH. All the protein in the fractions from 1-57 would be many fold greater than that in the "dimer range" from 57 to 64. Also, the lack of distinct peaks in the fractionation makes it look like the tetramer is just falling apart during preparation/fractionation. Maybe in the cells treated with H₂O₂, this tetramer is less stable for some reason? Can the authors also show some additional controls for the fractionation rather than just putting an arrow at the presumed size? Other nuclear proteins of defined molecular mass?
4. The effect of E7 expression on H3K79 methylation is very modest. Has this been quantitated? Has it been quantitated at the promoter of the genes shown in figure 4?
5. How do the effects of H₂O₂ on gene expression change in the absence of "normal" antioxidant response? (ie NRF2 knockout?). Is this redundant or additive response?
6. What is the concentration of aHB in the nucleus of cells with E7 expression?
7. What is your "unpublished proteomics data of HPV-positive and HPV-negative cervical cancer tissues identified LDH-A as the most significant protein linked to ROS homeostasis" lines 46-47?

Reviewer #3:

Remarks to the Author:

It is difficult to understand why the authors have not described the proteomic results with respect to HPV positive and HPV negative cancer tissue (see page 3 lines 45-47).

The title as stated is "Nuclear LDH-A Produces 1 α -Hydroxybutyrate to Activate Antioxidant Responses 2 and Promote HPV-induced Cervical Carcinoma"; but contrary to the title, the authors actually do NOT detail HPV-induced cervical carcinoma development and promotion.

My impression is that the authors aimed to show HPV's and LDH's role in cancer development, but alternatively studied the HPV and LDH links in established cancer tissue and cells. Why did the authors not study non-neoplastic primary cells (i.e. organoids)? Instead, they appear to have used HPV-negative cancer cell lines (e.g. HT-3) which would have resulted in a plethora of epigenetic and genetic alterations and transfected these with HPV16E7. Furthermore, they also utilised HeLa cells, an HPV pos cell line, which have been in existence for many decades.

Had the authors used normal (non-neoplastic) primary cells for the experiments illustrated in Suppl Figure 3, then it is more than likely that an entirely different pattern would have evolved. For example, it is well known that H3K27me3 and subsequent DNA methylation of polycomb-group target genes play a crucial role in cancer development and all of these epigenetic effects would have already occurred in the established cancer cells which the authors used and hence would not have been observed upon further manipulation of these cancer cells.

Human cervical carcinogenesis would be relatively easy to study given the availability of cells and tissues (within cervical cancer screening and resulting colposcopy referrals) in the pre-invasive setting.

It is unclear why in Figure 3c that, quite unexpectedly, a different cell line (i.e. HeLa) was employed which is HPV positive.

It is unclear how the target genes in Figure 4e were chosen.

With respect to Figure 3b, it is not immediately obvious what the top two and the bottom two panels represent...only after some time did I realise that the bottom two panels are a continuation of the top two panels.

The effects described in Figures 4c, 4d and 4e are somewhat unclear and I am unsure as to the authors' use of words such as "significantly", etc.. The proposed link between LDH-A and DOTL1 and H3K79me3 is not sufficiently supported by the data provided.

The clinical data provided in Figure 6 and in Supplementary Figure 5 are rather irrelevant for cervical carcinogenesis because the authors analysed only cancer tissue and correlated HPV17E7 scores with H3K79me3 scores. In addition, the description of IHC in Mat/Meth does not match with the results provided (i.e. in Mat/Meth only scores 0, 1, 2 and 3 were provided but in the relevant Figures the scores increased to 12).

Point by point response to reviewers' comments

Reviewer #1:

In this manuscript, the authors provide evidence that ROS induction or expression of the oncoprotein E7 from the mucosal high (HR) human papillomavirus type 16 (HPV16) induces nuclear translocation of the dimeric form of lactate dehydrogenase A (LDHA). This nuclear form of LDHA promotes the accumulation of α -hydroxybutyrate (α -HB) via a non-canonical activity. High nuclear levels of α -HB lead to histone H3 Lys79 hypermethylation, resulting in the activation of genes encoding proteins involved in the antioxidant responses and Wnt signalling pathway.

Overall, the experiments are well performed. The authors' conclusions concerning the link between HPV carcinogenesis and LDHA are less convincing and require further work and clarifications as described in detail below. In addition, no experiments were performed in primary human keratinocytes that are naturally infected by the different HPV types.

Response: We thank the reviewer for the positive comments and kind suggestions. As the reviewer suggested, primary human cervix keratinocytes (PHKs) and HaCaT cells, an immortalized human keratinocytes cell line, were used as models in our study. First, we tested if HPV16/18 E7 expression could induce LDHA nuclear translocation through upregulation of cellular ROS level in PHKs. Consistent with the data obtained from HT-3 cells, intracellular ROS level was increased (new Fig. 1e) accompanied by the elevated percentage of LDHA nuclear translocated cells upon HPV16/18 E7 expression (new Fig. 1c, d). Next, LDHA enzyme activity assay was performed in HaCaT cells with or without HPV16 E7 expression. The non-canonical enzyme activity (α -HB-producing), but not the canonical activity (lactate-producing) of LDHA, was significantly elevated after HPV16 E7 expression (new Fig. 2b). Then, we validated the effect of HPV16/18 E7 on histone H3K79 methylation levels in PHKs and HaCaT cells. HPV16/18 E7 expression upregulated H3K79 methylation levels which could be partially blocked by NAC (new Fig. 4c). Moreover, the binding of LDHA and DOT1L was enhanced in HaCaT cells upon HPV16/18 E7 expression (new Supplementary Fig. 5d). Together, these data strongly validate our hypothetical model in cervical non-neoplastic primary cells.

new Fig. 1c

new Fig. 1d

new Fig. 1e

new Fig. 2b

new Fig. 4c

new Supplementary Fig. 5d

Legend: H3K79me1 H3K79me2 H3K79me3

Specific points:

(1) Results, lines 48-50: “we collected HPV-positive (n = 38) and HPV-negative (n = 28) cervical cancer specimens from 66 patients”. It is well known that the majority of cervical cancers are positive for HPV16 or other 11 HR HPV types, e.g. 18, 31, 33, 35 45, etc. Therefore, it is unlikely that the authors managed to identify 28 HPV-negative cervical cancers. No information is provided about the characterization of the 66 cervical cancers. Did the authors check for the presence of DNA and RNA of the different HR HPV types? This analysis is essential for the classification of HPV-negative or HPV-positive cervical cancers. Without this additional work, the interpretation of the results shown in Figure 1 and 6 is impossible.

Response: As the reviewer mentioned, we checked the 66 cervical cancer specimens for the presence of 12 different HR-HPV types (HPV-16, 18, 31, 33, 35, 39, 45, 51, 52, 56, 58, 59). The HR-HPV information of the samples was provided below (new Supplementary Table 2).

new Supplementary Table 2. HR HPV-type characterization of 66 cervical specimens

Sample NO.	HR-HPV types	Sample NO.	HR-HPV types
HPV(-)-1	None	HPV(+)-1	16
HPV(-)-2	None	HPV(+)-2	16
HPV(-)-3	None	HPV(+)-3	16
HPV(-)-4	None	HPV(+)-4	16,35
HPV(-)-5	None	HPV(+)-5	16
HPV(-)-6	None	HPV(+)-6	16
HPV(-)-7	None	HPV(+)-7	11,16
HPV(-)-8	None	HPV(+)-8	16
HPV(-)-9	None	HPV(+)-9	16
HPV(-)-10	None	HPV(+)-10	16,18
HPV(-)-11	None	HPV(+)-11	16
HPV(-)-12	None	HPV(+)-12	16
HPV(-)-13	None	HPV(+)-13	16
HPV(-)-14	None	HPV(+)-14	16,51
HPV(-)-15	None	HPV(+)-15	16
HPV(-)-16	None	HPV(+)-16	16

HPV(-)-17	None	HPV(+)-17	16,18
HPV(-)-18	None	HPV(+)-18	16
HPV(-)-19	None	HPV(+)-19	16
HPV(-)-20	None	HPV(+)-20	16
HPV(-)-21	None	HPV(+)-21	16
HPV(-)-22	None	HPV(+)-22	16,45
HPV(-)-23	None	HPV(+)-23	16
HPV(-)-24	None	HPV(+)-24	16
HPV(-)-25	None	HPV(+)-25	16
HPV(-)-26	None	HPV(+)-26	16
HPV(-)-27	None	HPV(+)-27	16,52
		HPV(+)-28	16
		HPV(+)-29	16
		HPV(+)-30	16
		HPV(+)-31	16
		HPV(+)-32	16
		HPV(+)-33	16
		HPV(+)-34	16
		HPV(+)-35	16
		HPV(+)-36	16
		HPV(+)-37	16
		HPV(+)-38	16
		HPV(+)-39	18

(2) Figure 1c: More fields of the IF staining should be shown to better demonstrate the LDHA nuclear translocation in HPV16 E7 HT-3 cells.

Response: As the reviewer suggested, the IF staining pictures were given less magnified fields to better demonstrate the LDHA nuclear translocation in HT-3 cells with or without HPV16 E7 expression (new Supplementary Fig. 2b). In addition, PHKs were also used to test the effect of HPV16/18 E7 on LDHA nuclear translocation. The expression of HPV16/18 E7 significantly increased LDHA's nuclear translocation while NAC could also block the effect (new Fig. 1c, d).

new Supplementary Fig. 2b

new Fig. 1c

new Fig. 1d

(3) Figure 3c: HeLa cells express E6 and E7 from HR HPV type 18. Why is LDHA not in the nucleus of these cells? Is the ability to promote LDHA translocation into the nucleus a feature of HPV16, and not of HPV18 or other HR HPV types? Since the authors state throughout the manuscript that they have identified a key event in HPV-mediated carcinogenesis, a comparative analysis of the properties of the different HR HPV oncoproteins in promoting LDHA nuclear accumulation should be performed. Ideally, primary human keratinocytes should be used as an experimental model, since they represent the natural target of HPVs. Regarding the experiments performed in HeLa cells, without the demonstration that HPV18 E7 is not able to promote LDHA nuclear translocation, it is difficult to interpret the results obtained in HeLa cells shown in many Figures of the manuscript.

Response: During high-risk HPV infection, viral differentiation-dependent promoters become upregulated as characterized by the increased expression of two viral early genes, *E6* and *E7*, serving as the major initiator of cell transformation^{1, 2}. However, recent studies reported that HPV16 E7 was the more potent driver for cervical cancer^{3, 4, 5, 6, 7}, and elevation of HPV16 E7 was required to sustain a malignant phenotype in primary cervical cancer, even in the presence of E6 oncoprotein^{5, 8}. HPV16 E7 has also been shown as an essential factor for viral replication in human keratinocytes⁹. Given all that, exogenous expression of HPV16/18 (two most common HR-HPV types in clinical) E7 in primary human cervix keratinocytes (PHKs)^{10, 11}, immortalized human keratinocyte HaCaT cells and HT-3 cells were used to mimic the process of malignant transformation.

As the reviewer suggested, to test whether the ability to promote LDHA nuclear translocation was a feature of HPV16, we transfected HPV16 E7 or HPV18 E7 into PHKs. Both HPV16 E7 and HPV18 E7 increased intracellular ROS level and promoted LDHA nuclear translocation, indicating that the ability to promote LDHA nuclear translocation was more likely to be E7 driven (new Fig. 1c-e). Meanwhile, mentioned by the reviewer, HeLa cells have been reported to contain HPV18 sequences. Unexpectedly, most of LDHA were located at cytoplasm but not nucleus under normal condition (new Supplementary Fig. 2g). In our opinion, the nuclear translocation of LDHA should be a demand against the pressure of cell survival from an instant stress (such as virus infection) or a growth limitation in tumors rather than a permanent event in a well-conditioned

cultured cell line. LDHA protein level was not changed upon H₂O₂ treatment (new Fig. 1h, and 3a-Input panels), referring that LDHA's nuclear translocation decreased the pool of cytosolic LDHA (new Fig. 1f, and new Supplementary Fig. 2e, g), which played an indispensable role in glycolysis that majorly took place at cytosol. Given that there was no more good evidence to support this hypothesis, we performed the IF staining upon H₂O₂ treatment in HaCaT cells (new Fig. 1f), which was complementary to the related data in HeLa cells (new Supplementary Fig. 2e). And HaCaT cells replaced several experiments in HeLa cells.

new Fig. 1f

new Supplementary Fig. 2e

(4) Figure 3b: gel filtration fractionations using cytoplasmic and nuclear extracts should be added to corroborate their interpretation of the results.

Response: Following the reviewer's suggestion, traditional nuclear fractionation assays were performed to separate the cytoplasmic and nuclear fractions with or without H₂O₂ treatment in HT-3 cells. However, nuclear LDHA was not detectable in nuclear fractions. With the functional nuclear isolation (used for nuclear metabolites measurement) applied, the nuclear LDHA were readily detected using western blotting under the same condition (new Fig. 1h). These results suggest that nuclear LDHA may leaked out due to low molecular weight (37 kDa) during the traditional nuclear fractionation procedures. Whereas the high-cost functional isolated nuclei were too expensive to meet the demands for gel filtration with western blotting detection, instead, protein cross-linking assay was performed to better interpret the polymerization of nuclear LDHA. LDHA tetramers were dramatically decreased by H₂O₂ treatment, accompanied by increased dimer and monomer (new Fig. 3a). Furthermore, LDHA^{NLS} group presented much more dimer fractions than that of LDHA^{NES} group (new Fig. 3d, e). Taken together, these results demonstrate that LDHA presents dimer forms rather than tetramer forms in the nucleus.

(5) Figures 4a and b: quantification of the protein band in at least three independent experiments should be also shown.

Response: As the reviewer suggested, we added the quantification data of the western blot in three independent experiments (new Fig. 4a, b). Furthermore, the impact of α -HB or H₂O₂ treatment, and HPV16/18 E7 expression on H3K79 methylation levels were determined in HaCaT cells and PHKs. Histone H3K79 tri-methylation level was significantly increased upon α -HB (new Fig. 4a) or H₂O₂ treatment (new Fig. 4b) or HPV16/18 E7 induction (new Fig. 4c), which was partially reversed by NAC (new Fig. 4b, c).

(6) Figure 4e: The mechanism of the interaction LDHA and DOT1L should be better investigated. Is this interaction mediated by post-translation modifications of these proteins?

Response: As the reviewer suggested, we further investigated the mechanism of the interaction between LDHA and DOT1L. Upon HPV16/18 E7 expression, nuclear α -HB concentration was increased about two-folds. Along with, the treatment of α -HB to LDHA^{NES} cells with α -HB restored H3K79 tri-methylation, we hypothesized that α -HB produced by nuclear LDHA might take part in the activation of DOT1L through its binding to LDHA. To test this, we found that treatment of cells with α -HB alone increased the binding between endogenous DOT1L and LDHA (new Fig. 4g).

Next, we further investigated the post-translational modifications of LDHA or DOT1L on their binding. Protein phosphorylation was identified in LDHA at multiple sites regulating their functions. To test whether the phosphorylation of LDHA was participated in its binding with DOT1L, specific protein tyrosine phosphatase inhibitor, etidronate, and protein serine/threonine phosphatase inhibitor, calyculin A, were used to stimulate LDHA phosphorylation and then co-immunoprecipitations were performed. Etidronate, but not calyculin A, strongly increased the binding between LDHA and DOT1L (Fig. a). Unexpectedly, LDHA tyrosine phosphorylation level was not increased upon etidronate treatment (repeated three times independently). To this end, we performed co-immunoprecipitations (Co-IP) to test if DOT1L was tyrosine-phosphorylated and contributed to their binding. However, no tyrosine phosphorylation was detected on DOT1L (Fig. b). These data indicate that there may exist other protein participating the binding between LDHA and DOT1L through its tyrosine phosphorylation, which needs further investigation.

Meanwhile, LDHA acetylation at lysine 5 (K5) was reported to play an important role in tumor growth and cancer metabolism¹². To better investigate whether the interaction was mediated or inhibited by LDHA acetylation, we used protein deacetylase inhibitor NAM to stimulate LDHA K5 acetylation level (detected using LDHA K5-ac specific antibody) and found that NAM treatment dramatically decreased the binding between LDHA and DOT1L with raised LDHA K5 acetylation level (Fig. c). Similarly, LDHA K5-ac level was negatively correlated with its interaction with DOT1L upon α -HB treatment (new Fig. 4g lower panel with K5-ac added). These data suggest that LDHA K5 acetylation aborts the binding between LDHA and DOT1L.

new Fig. 4g

Fig. c

Fig. a

Fig. b

new Fig. 4g lower panel with K5-ac

Minor points

(1) Abstract, lines 4-5: “High-risk human papilloma virus (HPV) infection is the main cause of cervical cancer, however the underlying mechanism remains poorly defined”. This statement is incorrect. In the last 20 years, an enormous number of studies have elucidated many mechanisms of E6 and E7 oncoproteins involved in cellular transformation. The authors could check some of the most recent reviews on HPV and carcinogenesis.

Response: We thank the reviewer for the kind suggestion and we modified the sentence in the Abstract as follow: It is well known that high-risk human papilloma virus (HR-HPV) is strongly associated with cervical cancer. Although tremendous progress has been made on the field, it remains to provide new insight into the underlying mechanism.

(2) Abstract: please give the definition of DOT1-like (Disruptor of telomeric silencing 1-like).

Response: The definition of DOT1L was added in the manuscript: Surprisingly, nuclear LDHA retained its classic activity but gained a non-canonical enzyme activity to produce α -hydroxybutyrate and triggered DOT1L (disruptor of telomeric silencing 1-like)-mediated histone H3 Lys79 hypermethylation, resulting in the activation of antioxidant responses and Wnt signaling pathway.

(3) Introduction: line 29 “...but the underlying mechanism remains undefined.” Please see point 1 above.

Response: The first paragraph of Introduction is revised as follow:

Cervical cancer is the third most common cancer in women worldwide with about 528,000 new cases and 266,000 deaths annually¹³. Among those, about 95% cases are caused by persistent infections with HR-HPVs¹⁴. During high-risk HPV infection, viral differentiation-dependent promoters become upregulated as characterized by the increased expression of two viral early genes, *E6* and *E7*, serving as the major initiator of cell transformation^{1,2}. Moreover, recent studies reported that HPV16 *E7* was the more potent driver for cervical cancer^{3,4,5,6,7}, and elevation of

HPV16 E7 was required to sustain a malignant phenotype in primary cervical cancer⁸. HPV16 E7 has been shown as an essential factor for viral replication in human keratinocytes⁹. And yet, the new mechanism of E7 protein on HPV-induced cervical carcinogenesis remains to be discovered.

(4) Introduction, lines 24-26: “During high-risk HPV infection, viral differentiation-dependent promoters become upregulated that is characterized by the increased expression of two viral early genes, E6 and E7”. As far as I know, only the activity of the late promoter, which controls L1 and L2 expression, is regulated in a differentiation-dependent manner. The accumulation of E6 and E7 in the suprabasal layers of the infected epithelium is mainly explained by a high HPV genome amplification.

Response: The first paragraph of Introduction is revised as follow:

Cervical cancer is the third most common cancer in women worldwide with about 528,000 new cases and 266,000 deaths annually¹³. Among those, about 95% cases are caused by persistent infections with HR-HPVs¹⁴. During high-risk HPV infection, viral differentiation-dependent promoters become upregulated as characterized by the increased expression of two viral early genes, *E6* and *E7*, serving as the major initiator of cell transformation^{1,2}. Moreover, recent studies reported that HPV16 E7 was the more potent driver for cervical cancer^{3,4,5,6,7}, and elevation of HPV16 E7 was required to sustain a malignant phenotype in primary cervical cancer⁸. HPV16 E7 has been shown as an essential factor for viral replication in human keratinocytes⁹. And yet, the new mechanism of E7 protein on HPV-induced cervical carcinogenesis remains to be discovered.

(5) Legends Figure 3b and d should provide a short explanation for the numbers “158” and “43”, which most likely represent the molecular weight markers.

Response: As the reviewer suggested, the explanation for the number 158 and 43 is added in the legends for new Fig. 3b as follow: Molecular-mass, 158 and 43 kDa marked below the blots, were determined by Gel Filtration Calibration Kit HMW (GE Healthcare).

(6) Line 128: The sentence refers to Figure 3e and not 3d.

Response: We have revised it in our new manuscript.

(7) Reference 8 is not complete

Response: We have revised it in our new manuscript.

Reviewer #2:

Nuclear LDHA Produces α -Hydroxybutyrate to Activate Antioxidant Responses and Promote HPV-induced Cervical Carcinoma

The authors present evidence supporting a model in which infection of epithelial cells with high risk HPV results in a burst of reactive oxygen species, translocation of LDH to the nucleus and activation of a gene profile that supports the growth of cervical cancer. The experiments appear to be well done as far as they go, but the overall model has some significant gaps in logic and experimental support. (1) First of all, the introduction of HPV E7 into untransformed cervical cells is not equivalent to infection with replication competent virus. (2) Therefore, data is necessary to support the conclusion that E7 alone can induce ROS. How does this occur? Is there altered mitochondrial function? P450 activation? (3) And if this is such a stressful event, why does it not activate the more conventional KEAP-NRF pathway to detoxify the species, rather than invoking a whole new molecular mechanism?

Response: We thank the reviewer for the positive comments and kind suggestions.

(1) Recent studies reported that HPV16 E7 was the potent driver for cervical cancer^{3, 4, 5, 6, 7}, and elevation of HPV16 E7 was required to sustain a malignant phenotype in primary cervical cancer, even in the presence of E6 oncoprotein^{5, 8}. HPV16 E7 has also been shown as an essential factor for viral replication in human keratinocytes⁹. Given all that, exogenous expression of HPV16/18 (two most common HR-HPV types in clinical) E7 in primary human cervix keratinocytes (PHKs)^{10, 11}, immortalized human keratinocyte HaCaT cells and HT-3 cells were used to better mimic the process of cervical cells malignant transformation. First, we test if HPV16/18 E7 expression could induce LDHA nuclear translocation through upregulation of cellular ROS level in PHKs. Consistent with the data obtained from HT-3 cells, intracellular ROS level was increased (new Fig. 1e) accompanied by the elevated percentage of LDHA nuclear translocated cells upon HPV16/18 E7 expression (new Fig. 1c, d). Next, LDHA enzyme activity assay was performed in HaCaT cells with or without HPV16 E7 expression. The non-canonical enzyme activity (α -HB-producing), but not the canonical activity (lactate-producing) of LDHA, was significantly elevated after HPV16 E7 expression (new Fig. 2b). Then, we validated the effect of HPV16/18 E7

on histone H3K79 methylation levels in PHKs and HaCaT cells. HPV16/18 E7 expression upregulated H3K79 methylation levels which could be partially blocked by NAC (new Fig. 4c). Moreover, the binding of LDHA and DOT1L were enhanced in HaCaT cells upon HPV16/18 E7 expression (new Supplementary Fig. 5d). Together, these data validate our hypothetical model in cervical non-neoplastic primary cells.

(2) As the reviewer suggested, we further investigated the main source of the elevated ROS level. Using the mitochondrial superoxide indicator, we found that HPV16/18 E7 expression increased mitochondrial ROS level in HaCaT cells (new Supplementary Fig. 2j). Meanwhile, cells treated with diphenyleneiodonium (DPI), NADPH oxidase (NOX) inhibitor, showed a decreased ROS levels compared to control cells (new Supplementary Fig. 2k), suggesting that NOXs, as well as mitochondrial respiratory chain, contributes to HPV16/18 E7-induced intracellular ROS elevation.

new Supplementary Fig. 2j

new Supplementary Fig. 2k

(3) We found that the crucial antioxidant NRF2 pathway was activated upon HPV16/18 E7 induction in our study (new Supplementary Fig. 6f). However, with the treatment of EPZ004777, an inhibitor of DOT1L, the NRF2 activation is dramatically decreased (new Fig. 5c), indicating that HPV16/18 E7 induced H3K79 hypermethylation was necessary for NRF2 activation. On the other hand, *NQO1* and *GCLC*, two NRF2 target genes, were no longer activated upon H₂O₂ treatment in *NRF2* KO cells (new Fig. 5d) and NRF2 inhibitor treatment in HaCaT cells with HPV16/18 E7 expression (new Fig. 5e and new Supplementary Fig. 6h), suggesting that NRF2 is essential for E7-induced antioxidant response. Furthermore, in *LDHA* KO and *LDHA/NRF2* double knockout (DKO) cells, H₂O₂ treatment did not activate *NQO1* and *GCLC* gene expressions (new Supplementary Fig. 6k) indicating that both LDHA and NRF2 took indispensable roles in HPV16/18 E7-induced antioxidant response.

new Supplementary Fig. 6f

new Fig. 5c

new Fig. 5d

new Fig. 5e

new Supplementary Fig. 6h

new Supplementary Fig. 6k

If the reader accepts that LDH does translocate to the nucleus after E7 expression, then the easy explanation of its effect there would incorporate some level of redox signaling involving altered NAD⁺/NADH ratios. As an enzyme that typically consumes NADH while converting pyruvate to lactate, this would significantly decrease antioxidant pools in the nucleus, allowing for increased oxidative stress. (1) This would seem much more likely to alter gene expression (as several transcription factors have been shown to be redox sensitive (for review Cellular Signaling Volume 14, Issue 11, November 2002, Pages 879-897)). (2) The use of NAC as proof of mechanism is not really that conclusive because NAC will influence all redox balances in the cell (even nuclear levels of NAD⁺/NADH). (3) Additionally, the effects reported on H3K79 hypermethylation need to be more closely tied to changes in the expression of the genes listed in figure 4f.

Response: (1) As the reviewer suggested, we test the status of two other redox sensitive transcription factors, HIF-1 α and NF- κ B. NRF2, HIF-1 α , and NF- κ B were activated upon HPV16 E7 induction or H₂O₂ treatment (new Fig. 5c and Supplementary Fig. 6e). However, only the activation of NRF2 was induced by H3K79 hypermethylation (new Fig. 5c and Supplementary Fig. 6e). These data suggest that the activation of HIF-1 α and NF- κ B is independent on H3K79 methylation levels. Furthermore, in *LDHA* KO cells, the activation of NRF2 induced by H₂O₂ is attenuated (new Fig. 5f), indicating that ROS-induced NRF2 activation was partially dependent on LDHA while ROS-induced HIF-1 α or NF- κ B activation was independent on LDHA (new Supplementary Fig. 6i). These data suggest that HPV16 E7 or H₂O₂-induced activation of NRF2, but not HIF-1 α or NF- κ B, is LDHA-dependent.

new Fig. 5c

new Supplementary Fig. 6e

new Fig. 5f

new Supplementary Fig. 6i

(2) In our study, we discovered that LDHA sensed ROS to upregulate H3K79 methylation levels and thus promoted antioxidant and Wnt target gene expression. NAC (N-acetyl-L-cysteine) contains a free thiol group, which is able to reduce free radicals, and functions as a ROS scavenger. Indeed, the supplement of NAC scavenges all sources of ROS, including the HPV16/18 E7-induced, H₂O₂ treatment-induced, and endogenous ROS which plays important roles in cellular signaling transduction. As no specific ROS scavenger that we could use for diminishing the HPV16/18 E7-induced or H₂O₂ treatment-induced ROS, we chose a relatively low concentration (1 mM) for NAC treatment to reduce its impact on endogenous ROS as far as possible.

(3) As the reviewer suggested, chromatin immunoprecipitations (ChIP)-qPCR were performed to verify the changes on the genes expression was the consequence of H3K79 hypermethylation. *SOD1*, *CAT*, *CTNNB1*, and *MYC* genes were more enriched using H3K79me2 antibody upon HPV16/18 E7 expression in HaCaT (new Fig. 5b and Supplementary Fig. 6c) and HT-3 cells (new Supplementary Fig. 6d).

new Supplementary Fig. 6c

new Supplementary Fig. 6d

Finally, can the authors show any significant effect of nuclear LDH on characteristics of cellular transformation? Does the expression of LDH^{nls} versus LDH^{nes} versus LDH^{wt} alter growth of tumor cells in nude mice? Can any increase be directly attributed to gene expression changes identified here?

Response: As the reviewer suggested, LDHA^{KO} with vector, LDHA^{WT}, LDHA^{NLS}, and LDHA^{NES} rescue cells were injected subcutaneously into nude mice. LDHA^{WT} group showed increased tumor growth compared to that of vector group, while LDHA^{NLS} but not LDHA^{NES} group displayed significantly increased tumor growth compared with that of other groups (new Fig. 7a-c). This result suggests that nuclear LDHA promotes tumor growth. In agreement, elevated NRF2,

SOD1, MYC and H3K79 tri-methylation levels were observed in LDHA^{NLS} xenografts (new Fig. 7d).

new Fig. 7a

new Fig. 7b

new Fig. 7c

new Fig. 7d

Major points:

1. In figure 1f, the other cellular fractions need to be run and compared to the nuclear fraction. How much of the LDH is nuclear? How much of the LDH translocates to the nucleus after E7 expression?

Response: As different nuclear fractionation protocols were performed, we found that nuclear LDHA may leak out due to low molecular weight (37 kDa) during the nuclear fractionation procedures: (1) Two different traditional nuclear fractionation assays were performed to separate the cytoplasmic and nuclear fractions with or without H₂O₂ treatment in HT-3 cells, and nuclear LDHA was not detectable in nuclear fractions; (2) With the functional nuclear isolation (used for nuclear metabolites measurement) applied, the nuclear LDHA were readily detected using western blotting under the same condition (new Fig. 1h). As the blots in new Fig. 1h were quantified, about 25% LDHA were translocated into nucleus (Fig. a). However, it was hard to say that the functional nuclear isolation protocol could separate the nuclei completely. Instead, to better quantify the status of LDHA nuclear translocation, LDHA nuclear translocated cells were quantified in five individual immunostained sections, and 500 or more cells were counted for each condition. In over 40% of the cells, LDHA were translocated into the nucleus upon HPV16/18 E7 expression (new Fig. 1c, d).

new Fig. 1c

new Fig. 1d

2. (1) What are the relative lactate producing versus aHB producing activities of LDH? All the reported activities appear to be normalized to sit is difficult to tell. Is the aHB producing activity half of the lactate producing activity? 1%? (2) How much extra aHB is produced by the increase the aHB activity of only 1.5 fold?

Response: (1) Based on our *in vitro* enzyme activity assay, the α -HB producing activity of LDHA was about 36.2% compared to its lactate producing activity under the same substrate concentration (2mM α -KB or pyruvate) (new Supplementary Fig. 3a). Meanwhile, LDHA^{NLS} was shown to be more active for α -HB producing with a lower Km value (new Supplementary Fig. 3b), while no significant change on lactate producing ability (new Supplementary Fig. 3c).

new Supplementary Fig. 3a

new Supplementary Fig. 3b

	WT	NLS	NES
Vmax	1253	1804	957.1
Km	1.807	1.026	1.638

new Supplementary Fig. 3c

(2) As the reviewer mentioned, *LDHA* KO cells were used to better understand how extra α -HB was produced by LDHA with the 1.5-fold α -HB producing activity change. The standard curve was determined using the linear regression of the peak area of sodium α -hydroxybutyrate against the interval standards (new Supplementary Fig. 3d). Cellular α -HB concentration increased from 8.59 μ M to 15.58 μ M produced by LDHA. However, there was no α -HB increase in response to H_2O_2 treatment when absence of LDHA (new Supplementary Fig. 3i). These data indicate that the 1.8-fold accumulation of α -HB is produced by LDHA.

new Supplementary Fig. 3d

new Supplementary Fig. 3i

3. The biochemical fractionation of dimeric versus tetrameric LDH is totally unconvincing. Why did then authors only show fractions 43-68? Does fraction 43 represent the flow through? If the monomer is 36kDa, then the dimer is 70 kDa. There appears to be some LDH in this region between 43 and 168 in the H_2O_2 treated cells, but this looks to be a very small fraction of the total LDH. All the protein in the fractions from 1-57 would be many fold greater than that in the “dimer range” from 57 to 64. Also, the lack of distinct peaks in the fractionation makes it look like the tetramer is just falling apart during preparation/fractionation. Maybe in the cells treated with H_2O_2 , this tetramer is less stable for some reason? Can the authors also show some additional controls for the fractionation rather than just putting an arrow at the presumed size? Other nuclear proteins of defined molecular mass?

Response: First, the gel filtration molecular mass was determined by recombinant thyroglobulin (669 kDa, refer to Fraction #40~#41), aldolase (158 kDa, refer to Fraction #56~#57), and ovalbumin (43 kDa, refer to Fraction #64~#65) (new Supplementary Fig. 4a). During the gel

filtration process, the samples were injected (Fraction #0) and eluted by PBS with a 0.4 ml/min flow rate. Fractions were then collected every 0.25 ml per tube for further analysis. The proteins were eluted and firstly detected at Fraction #34 using HT-3 whole cell extracts (new Supplementary Fig. 4b). Theoretically, LDHA's molecular weight is 37kDa, so that dimer fractions should be presented around Fraction #60 and tetramers should be presented around Fraction #57. However, LDHA was detected throughout multiple fractions, which may owe to the diverse complexes containing LDHA in whole cell extracts. Upon H₂O₂ treatment, LDHA shifted into lower molecular weight fractions (new Fig. 3b, lower panel). In addition, protein crosslinking assay was performed to validate our hypothesis. After crosslinking, LDHA tetramers were dramatically decreased by H₂O₂ treatment, accompanied by increased dimer and monomer (new Fig. 3a). Next, HEK293T cells with *LDHA* KO and *LDHA*^{WT}, *LDHA*^{NLS}, and *LDHA*^{NES} putback were used to perform gel filtration and protein crosslinking assays. *LDHA*^{NLS} group presented much more dimer fractions than that of *LDHA*^{NES} group (new Fig. 3d, e, and Supplementary Fig. 4c, d). Taken together, these results demonstrate that ROS disrupt LDHA tetramer formation, which may facilitate its nuclear translocation.

new Supplementary Fig. 4a

new Supplementary Fig. 4b

new Fig. 3a

new Fig. 3b

new Supplementary Fig. 4c

new Fig. 3d

new Supplementary Fig. 4d

new Fig. 3e

4. (1) The effect of E7 expression on H3K79 methylation is very modest. Has this been quantitated? (2) Has it been quantitated at the promoter of the genes shown in figure 4?

Response: (1) As the reviewer suggested, we added the quantification data of three independent experiments on H3K79 methylation upon HPV16/18 E7 expression in PHKs, HaCaT, and HT-3 cells (new Fig. 4c and Supplementary Fig. 5b).

new Fig. 4c

new Supplementary Fig. 5b

(2) As the reviewer suggested, chromatin immunoprecipitations (ChIP)-qPCR were performed to verify the changes on the genes expression is the consequence of H3K79 hypermethylation. *SOD1*, *CAT*, *CTNNB1*, and *MYC* genes were more enriched using H3K79me2 antibody upon HPV16/18 E7 expression in HaCaT (new Fig. 5b and Supplementary Fig. 6c) and HT-3 cells (new Supplementary Fig. 6d).

new Fig. 5b

new Supplementary Fig. 6c

new Supplementary Fig. 6d

5. How do the effects of H₂O₂ on gene expression change in the absence of “normal” antioxidant response? (ie NRF2 knockout?). Is this redundant or additive response?

Response: As the reviewer suggested, qPCR and ChIP-qPCR assays were performed to study the effects of H₂O₂ on gene expression change in the presence or absence of the key antioxidant regulator NRF2. We found that increased mRNA expression of *NQO1* and *GCLC* was dramatically impaired by deletion of NRF2 or NRF2-specific inhibitor ML385 (new Fig. 5d, e and Supplementary Fig. 6h). Furthermore, *LDHA* KO significantly reduced *NQO1* and *GCLC* expression upon H₂O₂ treatment (new Fig. 5f). In addition, activation of antioxidant genes was markedly decreased in DKO cells (Supplementary Fig. 6k). Genetic ablation of *LDHA* remarkably blocked the increase of H3K79 di-methylation level of *SOD1*, *CAT*, *CTNNB1*, and *MYC*, while

deletion of NRF2 blocked the increase of H3K79 di-methylation level of *SOD1* and *CAT*, but not *CTNNB1* and *MYC* gene body (new Fig. 5g). Together, these data indicates that LDHA and NRF2 are all required for activating antioxidant responses against HPV16/18 E7 expression and H₂O₂ treatment under our tested conditions.

new Fig. 5d

new Fig. 5e

new Supplementary Fig. 6h

new Fig. 5f

new Supplementary Fig. 6k

new Fig. 5g

6. What is the concentration of aHB in the nucleus of cells with E7 expression?

Response: As the reviewer suggested, the α -HB standard curve was prepared with different doses of sodium α -hydroxybutyrate (new Supplementary Fig. 3d). Upon HPV16/18 E7 transfection, the nuclear α -HB concentration elevated from 3.6 μ M to 7.0 μ M (HPV16 E7) or 6.6 μ M (HPV18 E7) in HaCaT cells, respectively (new Supplementary Fig. 3e). The nuclear α -HB concentration elevated from 2.3 μ M to 3.2 μ M, then decreased to 2.4 μ M with NAC supplement (new Supplementary Fig. 3f).

new Supplementary Fig. 3d

new Supplementary Fig. 3e

new Supplementary Fig. 3f

7. What is your "unpublished proteomics data of HPV-positive and HPV-negative cervical cancers identified LDHA as the most significant protein linked to ROS homeostasis" lines

Response: We have added our proteomics data in our manuscript (new Supplementary Fig. 1 and

Supplementary Table 1) and described as follow:

The altered cellular redox levels may cause reversible modifications on specific cysteine residues and affect the protein functions. To decipher the changes in the cysteine proteome on HR-HPV infection, we developed a sensitive and specific redox proteomics method using iodoacetyl tandem mass tag (iodoTMT) reagents composed of a sulfhydryl-reactive iodoacetyl group selective labeling sulfhydryl (-SH) groups and sets of isobaric isomers which could be differentiated by mass spectrometry (MS), enabling quantitation of the relative abundance of cysteine modifications (Supplementary Fig. 1a). To identify HPV-related redox-sensitive effectors, the cysteine proteomes were obtained from C33A (HPV negative), SiHa (containing HPV16 genome), and HeLa (containing HPV18 genome) cells (Supplementary Fig. 1b). The key glycolysis enzyme LDHA was identified to be a potential key regulator in HPV-induced cervical cancer development (Supplementary Table 1).

Reviewer #3:

It is difficult to understand why the authors have not described the proteomic results with respect to HPV positive and HPV negative cancer tissue (see page 3 lines 45-47).

Response: We have added our proteomics data in our manuscript (new Supplementary Fig. 1 and Supplementary Table 1) and described as follow:

The altered cellular redox levels may cause reversible modifications on specific cysteine residues and affect the protein functions. To decipher the changes in the cysteine proteome on HR-HPV infection, we developed a sensitive and specific redox proteomics method using iodoacetyl tandem mass tag (iodoTMT) reagents composed of a sulfhydryl-reactive iodoacetyl group selective labeling sulfhydryl (-SH) groups and sets of isobaric isomers which could be differentiated by mass spectrometry (MS), enabling quantitation of the relative abundance of cysteine modifications (Supplementary Fig. 1a). To identify HPV-related redox-sensitive effectors, the cysteine proteomes were obtained from C33A (HPV negative), SiHa (containing HPV16 genome), and HeLa (containing HPV18 genome) cells (Supplementary Fig. 1b). The key glycolysis enzyme LDHA was identified to be a potential key regulator in HPV-induced cervical cancer development (Supplementary Table 1).

new Supplementary Fig. 1b

The title as stated is “Nuclear LDHA Produces 1 α -Hydroxybutyrate to Activate Antioxidant Responses 2 and Promote HPV-induced Cervical Carcinoma”; but contrary to the title, the authors actually do NOT detail HPV-induced cervical carcinoma development and promotion.

Response: As the reviewer suggested, some *in vivo* data were added in our manuscript. (1) To examine the effect of nuclear LDHA in tumor growth, *LDHA* KO with vector, *LDHA*^{WT}, *LDHA*^{NLS}, and *LDHA*^{NES} rescue cells were injected subcutaneously into nude mice. Consistent with previous reports, *LDHA*^{WT} group showed increased tumor growth compared to that of vector group, while *LDHA*^{NLS} but not *LDHA*^{NES} group displayed significantly increased tumor growth compared with that of other groups (new Fig. 7a-c). This result suggests that nuclear LDHA promotes tumor growth. In agreement, elevated NRF2, SOD1, MYC and H3K79 tri-methylation levels were observed in *LDHA*^{NLS} xenografts (new Fig. 7d). (2) To examine the effect of HPV16-induced LDHA nuclear translocation *in vivo*, we adopted a K14-HPV16 transgenic mouse model with which direct cervical application of HPV16 E7-targeted transcription activator-like effector nucleases (TALENs) effectively mutated the *E7* oncogene, reduced viral DNA load, and restored retinoblastoma-associated protein (RB1) function. Along with the TALEN applied, the cervical epithelium of the K14-HPV16 mouse showed a gradual loss of E7 and a reduction of epithelial proliferation. Meanwhile, the gradual loss of LDHA nuclear translocation correlated well with the decrease of H3K79 tri-methylation level in serial sections (new Fig. 7e). Strikingly, on day 24 when E7 expression was almost undetectable, nuclear LDHA was eliminated and H3K79 tri-methylation levels were dramatically attenuated (new Fig. 7e). These *in vivo* data

strongly support that HPV16 E7 induces LDHA nuclear translocation and H3K79 hypermethylation.

Besides, to present our work more concisely, we changed our manuscript title to “Nuclear Lactate Dehydrogenase A Senses ROS to Produce α -Hydroxybutyrate for HPV-Induced Cervical Tumor Growth” for better understanding to the readers.

new Fig. 7a

new Fig. 7b

new Fig. 7c

new Fig. 7d

new Fig. 7e

My impression is that the authors aimed to show HPV's and LDH's role in cancer development, but alternatively studied the HPV and LDH links in established cancer tissue and cells. Why did the authors not study non-neoplastic primary cells (i.e. organoids)? Instead, they appear to have used HPV-negative cancer cell lines (e.g. HT-3) which would have resulted in a plethora of epigenetic and genetic alterations and transfected these with HPV16E7. Furthermore, they also utilised HeLa cells, an HPV pos cell line, which have been in existence for many decades.

Response: As the reviewer suggested, primary human cervix keratinocytes (PHKs) and HaCaT cells, an immortalized human keratinocytes cell line, were used as models in our study. First, we tested if HPV16/18 E7 expression could induce LDHA nuclear translocation through upregulation of cellular ROS level in PHKs. Consistent with the data obtained from HT-3 cells, intracellular ROS level was increased (new Fig. 1e) accompanied by the elevated percentage of LDHA nuclear translocated cells upon HPV16/18 E7 expression (new Fig. 1c, d). Next, LDHA enzyme activity assay was performed in HaCaT cells with or without HPV16 E7 expression. The non-canonical enzyme activity (α -HB-producing), but not the canonical activity (lactate-producing) of LDHA, was significantly elevated after HPV16 E7 expression (new Fig. 2b). Then, we validate the effect of HPV16/18 E7 on histone H3K79 methylation levels in PHKs and HaCaT cells. HPV16/18 E7 expression upregulated H3K79 methylation levels which could be partially blocked by NAC (new Fig. 4c). Moreover, the binding of LDHA and DOT1L were enhanced in HaCaT cells upon HPV16/18 E7 expression (new Supplementary Fig. 5d). Together, these data strongly validate our hypothetical model in cervical non-neoplastic primary cells.

Had the authors used normal (non-neoplastic) primary cells for the experiments illustrated in Suppl Figure 3, then it is more than likely that an entirely different pattern would have evolved. For example, it is well known that H3K27me3 and subsequent DNA methylation of polycomb-group target genes play a crucial role in cancer development and all of these epigenetic effects would have already occurred in the established cancer cells which the authors used and hence would not have been observed upon further manipulation of these cancer cells.

Response: As the reviewer suggested, immortalized human keratinocytes, HaCaT cells, were used to identify histone methylation markers. Similarly, α -HB treatment could increase H3K79me2 levels, rather than other histone methylation markers, in a dose and time dependent manner (new Supplementary Fig. 5a).

new Supplementary Fig. 5a

Human cervical carcinogenesis would be relatively easy to study given the availability of cells and tissues (within cervical cancer screening and resulting colposcopy referrals) in the pre-invasive setting.

Response: As the reviewer suggested, we collected 70 pre-invasive human cervical specimens, containing 4 normal, 21 cervical intraepithelial neoplasia (CIN) grade I, 23 CIN grade II, and 22 CIN grade III samples. However, all the CIN samples were HPV positive. Along with, LDHA were localized at the nucleus without any distinctions on different CIN stages. Meanwhile, LDHA was localized at the cytosol in normal samples with HPV negative, while LDHA was localized at the nucleus even in normal samples with HPV positive. These data suggest that the nuclear translocation of LDHA may play a role in the early stage of HPV-induced malignant transformation.

It is unclear why in Figure 3c that, quite unexpectedly, a different cell line (i.e. HeLa) was employed which is HPV positive.

Response: The figure is to confirm the relationship between LDHA dimerization and its nuclear translocation. To avoid the misunderstanding, we have changed the model to the HEK293T cells with *LDHA* KO and *LDHA*^{WT/NLS/NES} putback (new Fig. 3e). In addition, protein crosslinking assay was performed to validate the tetramer-to-dimer change (new Fig. 3d). Meanwhile, as the reviewer mentioned, HeLa cells have been reported to contain HPV18 sequences. Unexpectedly, most of LDHA were located at cytoplasm but not nucleus under normal condition (new Supplementary Fig. 2g). In our opinion, the nuclear translocation of LDHA should be a demand against the pressure of cell survival from an instant stress (such as virus infection) or a growth limitation in tumors rather than a permanent event in a well-conditioned cultured cell line. LDHA protein level was not changed upon H₂O₂ treatment (new Fig. 1h, and 3a-Input panels), referring that LDHA's nuclear translocation decreased the pool of cytosolic LDHA (new Fig. 1f, and new Supplementary Fig. 2e, g), which played an indispensable role in glycolysis that majorly took place at cytosol. Given that there was no more good evidence to support this hypothesis, we replaced the data with HEK293T *LDHA* KO and putback cells.

new Fig. 3e

new Fig. 3d

new Supplementary Fig. 2g

new Fig. 1h

new Fig. 3a-Input panels

new Fig. 1f

new Supplementary Fig. 2e

It is unclear how the target genes in Figure 4e were chosen.

Response: As the reviewer asked, we added some descriptions in our text as follow: High ROS levels are generally detrimental to cells, and the increased antioxidant capacity becomes vital to maintaining tumor development and cell survival. In addition, aberrant activation of Wnt/ β -catenin signaling pathway, which is frequently observed in human cervical cancer, promotes cell proliferation and tumor progression^{15, 16}. Interestingly, the activation of DOT1L has been reported to modulate Wnt target genes expression^{17, 18}. To test whether the H3K79 hypermethylation induced by HPV16 E7 expression or H₂O₂ treatment further regulated cellular antioxidant responses or cell proliferation, seven antioxidant genes and three Wnt target genes were subjected to qPCR analysis.

With respect to Figure 3b, it is not immediately obvious what the top two and the bottom two panels represent....only after some time did I realise that the bottom two panels are a continuation of the top two panels.

Response: We are sorry for not clearly presenting our data. We run the #55 and #56 fraction twice as a control to keep the two blotting membranes (Fraction #43~56 and Fraction #55~68) at an approximate exposure condition, so that it would be comparable between two membranes (new Fig. 3b).

The effects described in Figures 4c, 4d and 4e are somewhat unclear and I am unsure as to the authors' use of words such as "significantly", etc.. The proposed link between LDHA and DOTL1 and H3K79me3 is not sufficiently supported by the data provided.

Response: As the reviewer suggested, we added the quantification data for the blots. Histone H3K79 tri-methylation level was significantly increased over 1.5-fold upon HPV16/18 E7 induction, which was partially reversed by NAC (new Fig. 4c). Cells expressing LDHA^{NLS} (1.43-fold), but not LDHA^{NES} (0.91-fold), showed elevated H3K79 tri-methylation level (new Fig. 4d). More importantly, supplementation of α -HB to LDHA^{NES} cells remarkably recovered H3K79 tri-methylation level from 0.91 to 1.29 (new Fig. 4d), suggesting that the non-canonical enzyme activity of nuclear LDHA is required for H3K79 hypermethylation. Moreover, EPZ004777 treatment blocked LDHA^{NLS}-induced upregulation of H3K79 tri-methylation (0.24 to 0.21) (new Fig. 4d), indicating that DOT1L was necessary for nuclear LDHA to increase H3K79 methylation.

To explore the mechanism underlying nuclear LDHA-induced H3K79 hypermethylation, co-immunoprecipitation was performed to determine the interaction between DOT1L and LDHA upon HPV16 E7 transduction or H₂O₂ treatment. Notably, both HPV16 E7 transduction and H₂O₂ treatment strengthened the binding of DOT1L with LDHA, which was partially blocked by NAC supplement (new Fig. 4e, f).

The clinical data provided in Figure 6 and in Supplementary Figure 5 are rather irrelevant for cervical carcinogenesis because the authors analysed only cancer tissue and correlated HPV17E7 scores with H3K79me3 scores. In addition, the description of IHC in Mat/Meth does not match

with the results provided (i.e. in Mat/Meth only scores 0, 1, 2 and 3 were provided but in the relevant Figures the scores increased to 12).

Response: We are sorry for not clearly presenting our data. We modified it in METHODS on “Human cervical tumor samples and immunohistochemistry (IHC)” as follow: The DAB signal of the cytoplasmic and nuclear LDHA or the nuclear HPV16 E7 and H3K79me3 were measured in this study. The mean value of HPV16 E7 signal intensity of each case was categorized into the corresponding groups by the following scores: 0 (negative staining); 1 (weak staining); 2 (moderate staining); and 3 (strong staining). To quantify the distribution of the nuclear LDHA and H3K79me3 signal intensities on four HPV16 E7 intensity groups, the mean value of each case was categorized into score 0 to 12. Further analysis was based on the IHC scores of LDHA, HPV16 E7, and H3K79me3.

References

1. Doorbar J. Molecular biology of human papillomavirus infection and cervical cancer. *Clin Sci (Lond)* **110**, 525-541 (2006).
2. Stoler MH, Rhodes CR, Whitbeck A, Wolinsky SM, Chow LT, Broker TR. Human papillomavirus type 16 and 18 gene expression in cervical neoplasias. *Hum Pathol* **23**, 117-128 (1992).
3. Mirabello L, *et al.* HPV16 E7 Genetic Conservation Is Critical to Carcinogenesis. *Cell* **170**, 1164-1174 e1166 (2017).
4. Jabbar SF, Abrams L, Glick A, Lambert PF. Persistence of high-grade cervical dysplasia and cervical cancer requires the continuous expression of the human papillomavirus type 16 E7 oncogene. *Cancer Res* **69**, 4407-4414 (2009).
5. Jabbar SF, *et al.* Cervical cancers require the continuous expression of the human papillomavirus type 16 E7 oncoprotein even in the presence of the viral E6 oncoprotein. *Cancer Res* **72**, 4008-4016 (2012).
6. Riley RR, Duensing S, Brake T, Munger K, Lambert PF, Arbeit JM. Dissection of human papillomavirus E6 and E7 function in transgenic mouse models of cervical carcinogenesis. *Cancer Res* **63**, 4862-4871 (2003).
7. Roman A, Munger K. The papillomavirus E7 proteins. *Virology* **445**, 138-168 (2013).

8. Cancer Genome Atlas Research N, *et al.* Integrated genomic and molecular characterization of cervical cancer. *Nature* **543**, 378-384 (2017).
9. Flores ER, Allen-Hoffmann BL, Lee D, Lambert PF. The human papillomavirus type 16 E7 oncogene is required for the productive stage of the viral life cycle. *J Virol* **74**, 6622-6631 (2000).
10. Liu X, Dakic A, Zhang Y, Dai Y, Chen R, Schlegel R. HPV E6 protein interacts physically and functionally with the cellular telomerase complex. *Proc Natl Acad Sci U S A* **106**, 18780-18785 (2009).
11. Rincon-Orozco B, *et al.* Epigenetic silencing of interferon-kappa in human papillomavirus type 16-positive cells. *Cancer Res* **69**, 8718-8725 (2009).
12. Zhao D, *et al.* Lysine-5 acetylation negatively regulates lactate dehydrogenase A and is decreased in pancreatic cancer. *Cancer Cell* **23**, 464-476 (2013).
13. Ferlay J, *et al.* Cancer incidence and mortality worldwide: sources, methods and major patterns in GLOBOCAN 2012. *Int J Cancer* **136**, E359-386 (2015).
14. Schiffman M, Wentzensen N, Wacholder S, Kinney W, Gage JC, Castle PE. Human papillomavirus testing in the prevention of cervical cancer. *J Natl Cancer Inst* **103**, 368-383 (2011).
15. Uren A, *et al.* Activation of the canonical Wnt pathway during genital keratinocyte transformation: a model for cervical cancer progression. *Cancer Res* **65**, 6199-6206 (2005).
16. Ramachandran I, *et al.* Wnt inhibitory factor 1 induces apoptosis and inhibits cervical cancer growth, invasion and angiogenesis in vivo. *Oncogene* **31**, 2725-2737 (2012).
17. Mohan M, *et al.* Linking H3K79 trimethylation to Wnt signaling through a novel Dot1-containing complex (DotCom). *Genes Dev* **24**, 574-589 (2010).
18. Ho LL, Sinha A, Verzi M, Bernt KM, Armstrong SA, Shivdasani RA. DOT1L-mediated H3K79 methylation in chromatin is dispensable for Wnt pathway-specific and other intestinal epithelial functions. *Mol Cell Biol* **33**, 1735-1745 (2013).

Reviewers' Comments:

Reviewer #2:

Remarks to the Author:

Ok.

Reviewer #3:

Remarks to the Author:

the authors have made a substantial effort to improve the manuscript.

Reviewer #4:

Remarks to the Author:

In the revised manuscript "Nuclear Lactate Dehydrogenase A Senses ROS to Produce α -hydroxybutyrate for HPV-Induced Cervical Tumor Growth", the authors have responded strongly to many of the previous criticisms. Many of the experiments have now been performed in primary cervical keratinocytes (PHK), the natural host for HPV infection. They also use HaCat cells, spontaneously immortalized keratinocytes, over PHKs for some experiments, but the reasoning behind this is unclear. Importantly, they have expanded their studies to include HPV18 in addition to HPV16 E7. However, there are still some issues that need to be addressed to increase the impact of their findings.

(1) A previous criticism was that no information was provided as to the characterization of the HPV-positive (n = 38) and HPV-negative (n = 28) cervical cancer specimens collected from 66 patients. It was unclear if the authors checked for the presence of DNA and RNA of the different HR HPV types.

In the revision, the authors indicate that they checked the 66 cervical specimens for 12 high-risk types, but still there is no indication of how this was carried and whether this was analysis of DNA or RNA. In addition, the HPV16+ samples are stratified on the basis of E7 expression (by IHC) (Fig. 7F), and it is unclear if these are cervical cancer specimens, or represent CIN1, CIN2, CIN3 lesions as well as cervical cancer. This information should be included as well as whether LDHA localization differs based on the stage of progression to cancer.

(2) Concern was raised over the original Figure 3C showing that LDHA was not nuclear in HeLa cervical carcinoma cells, which express HPV18 E6 and E7. This suggested that LDHA nuclear localization may be specific for HPV16 E7. The authors have now shown that expression of HPV18 E7 alone can induce nuclear translocation of LDHA and ROS production in PHKs. The authors suggest that nuclear localization of LDHA represents a stress response, which accounts for the cytoplasmic localization in HeLa cells. In this case, one would expect LDHA localization to be cytoplasmic in other cervical cancer lines (e.g. SiHa, Caski). Has this been examined? In this regard, does LDHA remain nuclear in established E7-expressing PHKs, or is this an initial response to E7 expression? Furthermore, E7 is expressed with E6 in tumor cells, and it is unclear whether LDHA is nuclear in PHKs expressing both E6 and E7. It is important to determine if E6 impacts LDHA nuclear localization as this could also account for the HeLa cell phenotype.

(3) It was suggested that for Figure 3b: gel filtration fractionations using cytoplasmic and nuclear extracts should be added to corroborate their interpretation of the results.

Protein cross-linking experiments have been performed to show that HaCat cells exhibit nuclear dimeric rather than tetrameric forms of LDHA in response to H₂O₂, but they do not show this occurs in cells expressing E7. They also need to corroborate the IF results for nuclear LDHA in E7-

expressing cells using the nuclear metabolite protocol they use for HaCaT and HT-3 cells in response to H2O2.

Minor points.

(1) Abstract Lines 5,6- The authors have not addressed the criticism that the statement "High-risk human papilloma virus (HPV) infection is the main cause of cervical cancer, however the underlying mechanism remains poorly defined" is inaccurate. The sentence has been revised to "Although tremendous progress has been made on the field, it remains to provide new insight into the underlying mechanism". However, this statement still disregards the research over the past 20 years that has provided insight into E6 and E7 mechanisms of transformation.

(2) Introduction- Lines 26-29 "During high-risk HPV infection, viral differentiation-dependent promoters become upregulated as characterized by the increased expression of two viral early genes, E6 and E7, serving as the major initiator of cell transformation. This is incorrect. The differentiation-dependent promoter is located within the E7 ORF and drives expression of E1, E2, E4, E5, L1 and L2. E6 and E7 are expressed from the early promoter located upstream of E6. This promoter is not differentiation-dependent.

Point by point response to reviewers' comments

Reviewer #2:

Ok.

Response: We thank the reviewer's positive comments on our revised manuscript.

Reviewer #3:

The authors have made a substantial effort to improve the manuscript.

Response: We thank the reviewer's positive comments on our revision.

Reviewer #4:

In the revised manuscript "Nuclear Lactate Dehydrogenase A Senses ROS to Produce α -hydroxybutyrate for HPV-Induced Cervical Tumor Growth", the authors have responded strongly to many of the previous criticisms. Many of the experiments have now been performed in primary cervical keratinocytes (PHK), the natural host for HPV infection. They also use HaCat cells, spontaneously immortalized keratinocytes, over PHKs for some experiments, but the reasoning behind this is unclear. Importantly, they have expanded their studies to include HPV18 in addition to HPV16 E7. However, there are still some issues that need to be addressed to increase the impact of their findings.

Response: We thank the reviewer for the positive comments and kind suggestions. As the reviewer #1 suggested, primary human cervix keratinocytes (PHKs) were used as an experimental model to better imitate the process of HPV infection. The key experiments studying the effect of E7 expression on LDHA nuclear translocation, ROS production, and H3K79 methylation were performed in PHKs. Besides, to better mimic the process of HPV-induced cervical carcinogenesis, a spontaneously immortalized keratinocytes cell line, HaCaT, was added to present a better model in several experiments, including the study of E7 expression on LDHA nuclear translocation, ROS production, enzyme activity alteration, LDHA dimerization, H3K79 methylation, and the binding between LDHA and DOT1L.

Specific points:

(1) A previous criticism was that no information was provided as to the characterization of the HPV-positive (n = 38) and HPV-negative (n = 28) cervical cancer specimens collected from 66 patients. It was unclear if the authors checked for the presence of DNA and RNA of the different HR HPV types.

In the revision, the authors indicate that they checked the 66 cervical specimens for 12 high-risk types, but still there is no indication of how this was carried and whether this was analysis of DNA or RNA. In addition, the HPV16+ samples are stratified on the basis of E7 expression (by IHC) (Fig. 7F), and it is unclear if these are cervical cancer specimens, or represent CIN1, CIN2, CIN3 lesions as well as cervical cancer. This information should be included as well as whether LDHA localization differs based on the stage of progression to cancer.

Response: We thank the reviewer for kind suggestions. First, we added the HPV analysis procedures in METHODS as follow and labeled it in Blue in Manuscript.

Tissue DNA extraction and HPV detection

The tissue DNA was extracted from 66 cervical cancer specimens using QIAamp DNA Mini Kit (Qiagen, 51304) following the manufacturer's protocol. The tissues were mechanically disrupted and lysed with proteinase K at 56 °C overnight, and DNA was purified using QIAamp Mini spin column. PCR-reverse dot blot (PCR-RDB) assay were performed using HPV Genotyping Detection Kit (Yaneng Biotech, Shenzhen, China) according to the manufacturer's instructions. The PCR reaction was amplified under the following conditions: an initial 50°C for 15 min, 95°C for 10 min; 40 cycles of 94°C for 30 s, 42°C for 90 s, and 72°C for 30 s; and a final extension at 72°C for 5 min. The PCR products were immobilized onto a nitrocellulose membrane and hybridized with typing probes. HPV subtypes were determined by the positive point on the HPV genotype profile on the membrane. HPV-positive and negative controls were also included in every experiment.

Next, as Fig. 7f showed that HPV16 E7 levels, LDHA nuclear translocation and H3K79 tri-methylation were positively correlated with each other in human cervical tumor tissues. As the

reviewer suggested, we try to find out if there are any relationship between LDHA translocation and the stage of cervical cancer progression. We collected 70 pre-invasive human cervical specimens, containing 4 normal, 21 cervical intraepithelial neoplasia (CIN) grade I, 23 CIN grade II, and 22 CIN grade III samples. However, all the CIN samples we collected were HPV positive. Along with, LDHA were localized at the nucleus without any distinctions at different CIN stages. Meanwhile, LDHA was localized at the cytosol in normal samples with HPV negative, while LDHA was localized at the nucleus even in normal samples with HPV positive. These data suggest that the nuclear translocation of LDHA may play a role in the early stage of HPV-induced malignant transformation.

(2) Concern was raised over the original Figure 3C showing that LDHA was not nuclear in HeLa cervical carcinoma cells, which express HPV18 E6 and E7. This suggested that LDHA nuclear localization may specific for HPV16 E7. The authors have now shown that expression of HPV18 E7 alone can induce nuclear translocation of LDHA and ROS production in PHKs. The authors suggest that nuclear localization of LDHA represents a stress response, which accounts for the cytoplasmic localization in HeLa cells. In this case, one would expect LDHA localization to be

cytoplasmic in other cervical cancer lines (e.g. SiHa, Caski). Has this been examined? In this regard, does LDHA remain nuclear in established E7-expressing PHKs, or is this an initial response to E7 expression? Furthermore, E7 is expressed with E6 in tumor cells, and it is unclear whether LDHA is nuclear in PHKs expressing both E6 and E7. It is important to determine if E6 impacts LDHA nuclear localization as this could also account for the HeLa cell phenotype.

Response: We thank the reviewer for the kind suggestions. As we observed that, although HeLa cells have been reported to contain HPV18 sequences, the most of LDHA were located at cytoplasm but not nucleus in HeLa cells under normal condition. We think that the nuclear translocation of LDHA should be a demand against the pressure of cell survival from an instant stress (such as virus infection) or a growth limitation in tumors rather than a permanent event in a well-conditioned cultured cell line. As the reviewer suggested, to test the LDHA localization in SiHa (containing HPV16 sequences) and Ca Ski (containing HPV16 and HPV18 sequences) cells, we performed the IF staining and found that LDHA was localized at cytosol as same as HeLa cells (Scale bar, 10 μ m).

Furthermore, we try to investigate whether LDHA remains nuclear or just an initial response to E7-expression, but it is really hard to define this question in PHKs model. The primary human cervix keratinocytes used in our study were isolated from cervical epithelial, and they were not immortalized cell line (even proliferated less than 20 population doublings). As previously, we found that NAC supplement could reverse E7 or H₂O₂-induced LDHA nuclear translocation in PHKs, indicating that LDHA localization is regulated by cellular ROS levels. Although we failed to test whether LDHA remains localized in the nucleus or not after longer period of time, but LDHA's nuclear translocation is reversible and regulated by ROS.

As the reviewer suggested, to investigate the effect of E6 on E7-induced LDHA nuclear

translocation, we packaged retrovirus containing HPV16 *E6*, *E7*, and *E6&E7* gene and infected PHKs. As expected, the expression of HPV16 *E7* significantly promoted the nuclear translocation of LDHA, but expression of HPV16 *E6* did not. Moreover, co-expression of HPV16 *E6&E7* showed dramatic LDHA nuclear localization (scale bar, 10 μ m), indicating that the presence of *E6* do not impact *E7*-induced LDHA nuclear translocation under our test conditions.

(3) It was suggested that for Figure 3b: gel filtration fractionations using cytoplasmic and nuclear extracts should be added to corroborate their interpretation of the results.

Protein cross-linking experiments have been performed to show that HaCat cells exhibit nuclear dimeric rather than tetrameric forms of LDHA in response to H₂O₂, but they do not show this occurs in cells expressing *E7*. They also need to corroborate the IF results for nuclear LDHA in *E7*-expressing cells using the nuclear metabolite protocol they use for HaCaT and HT-3 cells in response to H₂O₂.

Response: We thank the reviewer for the kind suggestion. As we mentioned in the last revision: Following the reviewer's suggestion, traditional nuclear fractionation assays were performed to separate the cytoplasmic and nuclear fractions with or without H₂O₂ treatment in HT-3 cells. However, nuclear LDHA was not detectable in nuclear fractions. With the functional nuclear isolation (used for nuclear metabolites measurement) applied, the nuclear LDHA were readily detected using western blotting under the same condition. These results suggest that nuclear LDHA may leaked out due to low molecular weight (37 kDa) during the traditional nuclear fractionation procedures. Whereas the high-cost functional isolated nuclei were too expensive to

meet the demands for gel filtration with western blotting detection, instead, protein cross-linking assay was performed to better interpret the polymerization of nuclear LDHA.

Furthermore, as the reviewer suggested, protein cross-linking assays were performed in HT-3 and HaCaT cells with or without E7 expression. We found that the expression of HPV16 E7 increased LDHA dimer to 1.9-fold in HaCaT cells and 1.5-fold in HT-3 cells, respectively (new Supplementary Fig. 4a). Also, the IF staining results for isolated nuclear in E7-expressing cells were shown to better illustrate the metabolite quantification data (new Supplementary Fig.3d).

Minor points:

(1) Abstract Lines 5,6- The authors have not addressed the criticism that the statement “High-risk human papilloma virus (HPV) infection is the main cause of cervical cancer, however the underlying mechanism remains poorly defined” is inaccurate. The sentence has been revised to “Although tremendous progress has been made on the field, it remains to provide new insight into the underlying mechanism”. However, this statement still disregards the research over the past 20 years that has provided insight into E6 and E7 mechanisms of transformation.

Response: We thank the reviewer for the precise suggestions. We modified the sentence in the Abstract as follow: It is well known that high-risk human papilloma virus (HR-HPV) infection is strongly associated with cervical cancer and E7 was identified as one of the key initiators in HPV-mediated carcinogenesis.

(2) Introduction- Lines 26-29 “During high-risk HPV infection, viral differentiation-dependent promoters become upregulated as characterized by the increased expression of two viral early genes, E6 and E7, serving as the major initiator of cell transformation. This is incorrect. The differentiation-dependent promoter is located within the E7 ORF and drives expression of E1, E2, E4, E5, L1 and L2. E6 and E7 are expressed from the early promoter located upstream of E6. This promoter is not differentiation-dependent.

Response: We thank the reviewer for the precise suggestions. To avoid any misunderstanding by the readers, we have modified the sentence in the Introduction as follow: During high-risk HPV infection, two viral early genes, *E6* and *E7*, were identified to play key roles in carcinogenesis by regulating signaling pathways related to cellular transformation.

Reviewers' Comments:

Reviewer #4:

Remarks to the Author:

The authors have responded very nicely to the previous criticisms. I have no further revisions.